# OMG-LLaVA 👁️🌋: Bridging Image-level, Object-level, Pixel-level Reasoning and Understanding

**Tao Zhang[1], Xiangtai Li[2,4] †, Hao Fei[2], Haobo Yuan[3], Shengqiong Wu[2],**
**Shunping Ji[1], Chen Change Loy[3], Shuicheng Yan[2]**
[1]Wuhan University [2]Skywork AI [3]S-Lab, NTU [4]Bytedance
**Project page:** https://lxtgh.github.io/project/omg_llava
*Email: xiangtai94@gmail.com and zhang_tao@whu.edu.cn*

## Abstract

Current universal segmentation methods demonstrate strong capabilities in pixel-level image and video understanding. However, they lack reasoning abilities and cannot be controlled via text instructions. In contrast, large vision-language multimodal models exhibit powerful vision-based conversation and reasoning capabilities but lack pixel-level understanding and have difficulty accepting visual prompts for flexible user interaction. This paper proposes OMG-LLaVA, a new and elegant framework combining powerful pixel-level vision understanding with reasoning abilities. It can accept various visual and text prompts for flexible user interaction. Specifically, we use a universal segmentation method as the visual encoder, integrating image information, perception priors, and visual prompts into visual tokens provided to the LLM. The LLM is responsible for understanding the user's text instructions and providing text responses and pixel-level segmentation results based on the visual information. We propose perception prior embedding to better integrate perception priors with image features. OMG-LLaVA achieves image-level, object-level, and pixel-level reasoning and understanding in a single model, matching or surpassing the performance of specialized methods on multiple benchmarks. Rather than using LLM to connect each specialist, our work aims at end-to-end training on one encoder, one decoder, and one LLM. The code and model have been released for further research.

## 1  Introduction

With the development of transformer models [94; 6; 93; 40; 71; 92; 64; 126; 49; 87; 10; 19; 58], recent works in both natural language processing (NLP) and computer vision raise one common trend: adopting one unified model to solve multiple tasks. For example, large language models (LLMs) [93; 40; 92] adopt scale-up models to solve multiple NLP tasks and achieve better results than previous expert models. In vision, we have also seen a similar trend [19; 58; 100; 99; 47; 112], adopting one model to solve multiple tasks or sub-tasks, including detection, segmentation, video analysis, low-level vision, pose estimations, and more tasks. Different methods adopt different transformer designs, including visual-in-context learning [99; 100], unified decoder [19; 58], and unified tokenizer [86; 16; 58]. In summary, benefiting from the *scalability* and *flexibility* of the transformer, adopting one model for all tasks has made a great progress [19; 71; 72; 70; 126; 88; 87].

---

Work done when Tao Zhang is an intern at Skywork AI. †: Project leader. Corresponding author: Xiangtai Li and Shunping Ji.

38th Conference on Neural Information Processing Systems (NeurIPS 2024).

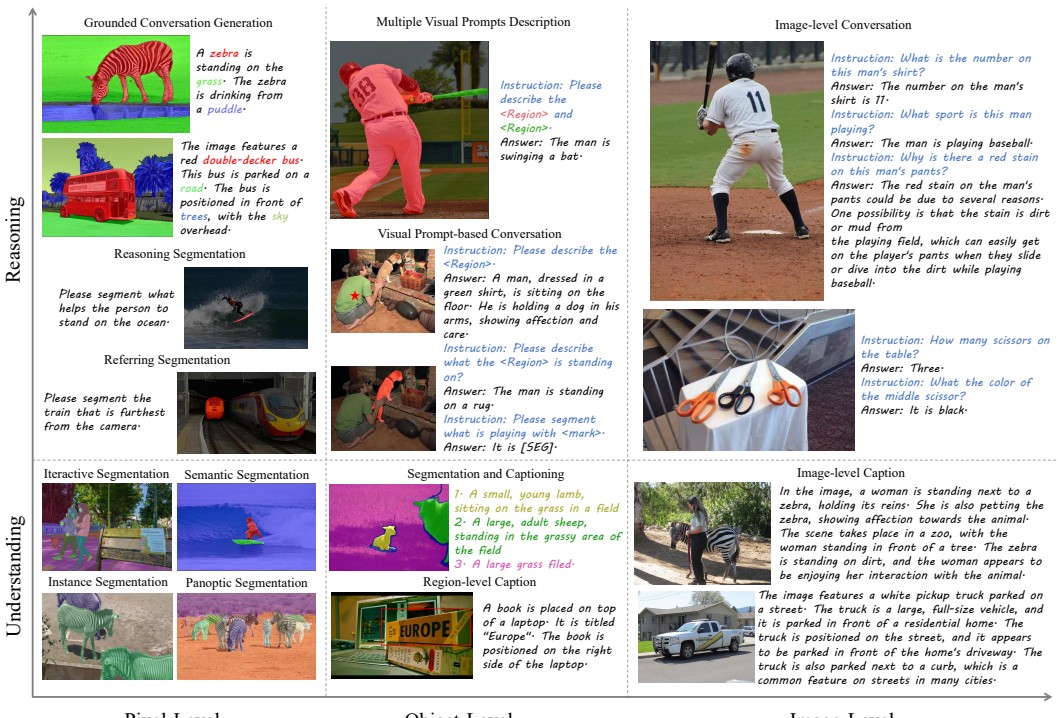

Figure 1: The comprehensive capabilities of OMG-LLaVA. OMG-LLaVA can handle a variety of pixel-level, object-level, and image-level understanding and reasoning tasks.

Meanwhile, by combining vision models and language models [71; 72; 70; 64; 65; 107], research on multi-modal models also adopts transformer-based design. One representative work, LLaVA [71; 72; 70], treats visual tokens as the inputs of LLMs and makes LLMs understand visual contents. Several works adopt similar designs [3; 13; 64; 18; 25], and all of them are termed Multi-modal Large Language Models (MLLMs). After that, most research focuses on improving MLLM benchmarks in various ways, including increasing data sizes [14; 18; 70] and enhancing the visual encoders [131; 24; 18] and visual resolutions [110; 18; 65; 25]. However, LLaVA-like models cannot output precise location information since they only carry out image-level analysis. Thus, recent works [126; 133; 11; 82; 13; 119; 128; 87; 67] try to fill this gaps by adding extra detection models for object level analysis, mask decoder for pixel-level analysis, visual prompts, and also propose task-specific instruction tuning with various datasets. By providing extra detection data and a decoder, the updated MLLMs can perform localization output. However, these models [135; 96; 49] are specifically tuned on specific tasks, losing the ability of LLaVA for image level analysis, such as caption and visual question answering. Meanwhile, several works [126; 49; 87; 78] adopt LLMs as agents to collaborate with various visual models or generation models. Despite the works being simple and effective, the inference and parameter costs are huge due to the multiple visual encoders and decoders. Moreover, there are no specific designs for task unification.

Motivated by the previous analysis, we ask one essential question: Can we bridge image-level, object-level, and pixel-level tasks into one MLLM model with only one LLM, one visual encoder, and one visual decoder? Back to the universal perception models, we can leverage these models to help us build a stronger MLLM to unify three-level inputs, including image, object, and pixel levels. In particular, we adopt OMG-Seg [58] as our universal perception model due to its simplicity and effectiveness in various segmentation tasks.

In this work, we present OMG-LLaVA, an elegant MLLM that bridges image-level, object-level, and pixel-level reasoning and understanding tasks in one model. We preserve the basic pixel-level segmentation ability of OMG-Seg by freezing the visual encoder and decoder, as shown in the bottom left of Fig. 1. Since the LLM processes text input, OMG-LLaVA can also perform referring segmentation, reasoning segmentation, and grounded conversation and generation, shown in the top left of Fig. 1. Moreover, as shown in Fig. 1, with the help of LLMs, OMG-LLaVA can also perform

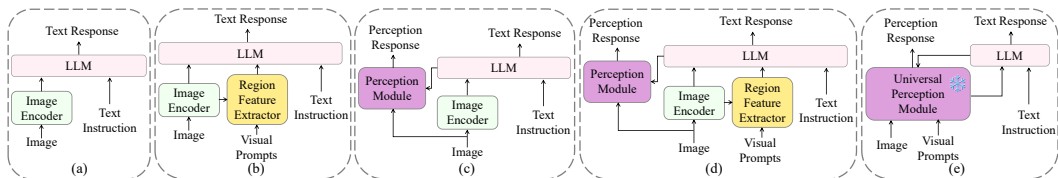

Figure 2: Summary of Current MLLM Architectures: (a) MLLMs with only image-level capability, including [71; 72; 70; 65], etc., (b) MLLMs with object-level capability, including [126; 87], (c) MLLMs with pixel-level capability, including [49; 88], etc., (d) MLLMs with both object-level and pixel-level capabilities but with a very complex system, such as [87], (e) OMG-LLaVA's architecture, which possesses an elegant and simple design while having image-level, object-level, and pixel-level capabilities.

image-level understanding as LLaVA, including caption and conversation, where most MLLMs for grounding lose such ability. In addition, OMG-LLaVA also supports the visual prompts as inputs, which results in object level understanding, such as visual prompt-based conversation and region-level captions. We achieve all these abilities using one LLM, one encoder, and one decoder.

In particular, to better encode the visual segmentation outputs, we propose a perception prior embedding module to absorb the object queries into object-centric visual tokens, which are the inputs of LLMs. We present a unified instruction formation strategy, which lets the model accept visual images, texts, and visual prompts as inputs and generate the response of text, segmentation tokens, segmentation masks, and labels. Following the LLaVA [71], we adopt pretraining and instruct tuning pipelines. Extensive experiments show the effectiveness of our components and training strategy. In addition to visual segmentation, OMG-LLaVA can also achieve good enough performance on 6 datasets, including COCO panoptic segmentation, VIPSeg video panoptic segmentation, refCOCO, refCOCO+, refCOCOg referring expression segmentation, GranDf grounded conversation generation, and refCOCOg region caption datasets. We hope our research can inspire the research on MLLM design in a more elegant way for the community.

## 2 Related Work

**Multimodal Large Language Models.** Early multimodal models [53] explore better fusion strategies, various feature extractors, and different meta-architectures. Most works focus on single tasks, such as caption and VQA. With the development of the large language models [6; 93; 40], recent works [52; 3; 92; 71; 17] mainly explore building an instruction-tuning pipeline for multiple multimodal benchmarks [39; 74; 62; 32]. LLaVA [71; 70; 69; 106; 136; 30; 28] is one earlier work that treats visual features as tokens. After that, several works [126] explore visual cues to enhance the visual inputs of LLaVA. On the other hand, several works [129; 127; 88; 131; 24; 25; 66; 130; 83; 38; 49] add extra components to adapt LLaVA for visual grounding, detection, segmentation, and video analysis. In particular, several works explore language-driven grounding and segmentation. However, these works are all trained with a specific purpose. We aim to build the simplest model to unify segmentation, instruction tuning, and prompt-driven segmentation in one model. To the best of our knowledge, we are the first model to achieve this goal.

**Unified Segmentation Models.** The vision transformers [10; 26; 79; 94] have led to research interest in universal segmentation. Recent works [95; 19; 123; 56; 21; 117; 115; 73; 91; 124; 122; 139; 111; 54; 141] have developed mask classification architectures with an end-to-end set prediction approach, outperforming previous specialized models [12; 46; 36; 57; 34; 60; 140] in both image, video and generalization segmentation tasks [45; 61; 59]. In particular, several works explore open-world segmentation, including entity segmentation [85; 84], open-vocabulary segmentation [125; 105]. Meanwhile, several works [58; 41; 111; 112; 33; 2] adopt one model with shared parameters to perform various segmentation tasks. One recent work, OMG-Seg [58], first unifies image, video, open-vocabulary, and interactive segmentation in one simple model. However, all of these works focus on visual segmentation and cannot generate interactive text and visual prompts, like MLLMs. Our work builds such a bridge to align MLLMs, visual segmentation, and prompt-driven segmentation models from joint co-training and model sharing, which serves as a new baseline for this field.

**Language-driven Location and Segmentation.** Early works [120; 68; 44; 23; 104; 135] in this direction mainly define the various language-driven tasks, including referring segmentation and

Table 1: Comparison of capabilities of different models. We include several representative methods here. Our OMG-LLaVA offers the most comprehensive capabilities, encompassing image-level, object-level, and pixel-level understanding and reasoning. Compared to [87; 35], OMG-LLaVA features an elegant and simple system architecture with only a single visual encoder.

| Method | Visual Encoder | Image-level | | Object-level | | | Pixel-level | | |
|---|---|---|---|---|---|---|---|---|---|
| | | Caption | Conversation | Visual Prompts | Caption | Conversation | Universal Seg | RES | GCG |
| LLAVA [71] | 1 | ✓ | ✓ | | | | | | |
| MiniGPT4 [142] | 1 | ✓ | ✓ | | | | | | |
| mPLUG-Owl [118] | 1 | ✓ | ✓ | | | | | | |
| LLaMA-Adapter [132] | 1 | ✓ | ✓ | | | | | | |
| Mini-Gemini [65] | 2 | ✓ | ✓ | | | | | | |
| InternVL 1.5 [18] | 1 | ✓ | ✓ | | | | | | |
| VisionLLM [97] | 1 | ✓ | ✓ | | | | | ✓ | |
| Shikra [13] | 1 | ✓ | ✓ | Point & Box | ✓ | ✓ | | | |
| Kosmos-2 [82] | 1 | ✓ | ✓ | Box | ✓ | ✓ | | | |
| GPT4RoI [133] | 1 | ✓ | ✓ | Box | ✓ | ✓ | | | |
| Ferret [119] | 1 | ✓ | ✓ | Point & Box & Mask | ✓ | ✓ | | | |
| Osprey [126] | 1 | ✓ | ✓ | Mask | ✓ | ✓ | | | |
| SPHINX-V [67] | 1 | ✓ | ✓ | Point & Box & Mask | ✓ | ✓ | | | |
| LISA [49] | 2 | ✓ | ✓ | | | | | ✓ | ✓ |
| GLAMM [87] | 2 | ✓ | ✓ | Box | ✓ | ✓ | | ✓ | ✓ |
| Groundhog [134] | 4 | ✓ | ✓ | Point & Box & Mask | ✓ | ✓ | | ✓ | ✓ |
| AnyRef [35] | 2 | ✓ | ✓ | Box | ✓ | ✓ | | ✓ | |
| PixelLM [88] | 1 | ✓ | ✓ | | | | | ✓ | |
| GSVA [109] | 2 | ✓ | ✓ | | | | | ✓ | |
| Groma [78] | 1 | ✓ | ✓ | Box | ✓ | ✓ | | | |
| VIP-LLaVA [8] | 1 | ✓ | ✓ | Point & Box & Mask | ✓ | ✓ | | | |
| PSALM [135] | 1 | | | Point & Box & Mask | | | ✓ | ✓ | |
| LaSagnA [102] | 2 | | | | | | | ✓ | |
| OMG-Seg [58] | 1 | | | Point | | | ✓ | | |
| OMG-LLaVA | 1 | ✓ | ✓ | Point & Box & Mask | ✓ | ✓ | ✓ | ✓ | ✓ |

referring localization. Most works [31; 5; 116; 77; 103; 105] design effective fusion modules to achieve better performance. Meanwhile, several works [55; 103; 108; 49; 87; 126; 81] explore more complex language-driven tasks from various aspects, including robustness, reasoning, and region-level caption. LISA [114] involves reasoning-based segmentation. Then, GLaMM [87] annotates a new dataset and proposes region-level caption and segmentation tasks. Meanwhile, several works [29; 72] use LLMs as agents to assign different visual experts. In contrast to these works, our method is a more elegant baseline, which contains **only** one visual encoder, one LLM, and one decoder.

**Visual Prompts.** With the prompting ability of LLMs, several works [100; 99; 4; 138; 90; 51; 81] also explore visual prompting methods in vision. According to the design and purposes, these works can be divided into different aspects, including learnable tokens [138], mask-visual-modeling for different tasks [100; 27; 98], and various visual prompting encoders for visual outputs [99; 101; 125; 47]. Our OMG-LLaVa also supports visual prompts for better interaction with the user's inputs, showing the potential for product purposes.

## 3 Methodology

### 3.1 Task Unification

**Motivation and Our Goals.** The LLMs unify most NLP tasks as token generation tasks and exhibit strong reasoning and instruction-following capabilities. As shown in Fig. 2 (a), LLaVA-like models [71; 70; 69; 110; 65; 131; 24; 25; 18; 64] further introduce visual tokens into LLMs, enabling LLMs to understand visual information and perform visual-based reasoning. However, they cannot accomplish fine-grained visual tasks like object-level and pixel-level understanding and reasoning. As shown in Fig. 2 (b), [126; 133; 11; 82; 13; 119] introduce region-level visual embeddings, allowing LLMs to achieve object-level understanding and reasoning tasks. However, these models rely on complex region embedding extraction designs. In addition, most cannot perform pixel-level understanding tasks. Thus, as shown in Fig. 2 (c),[49; 88; 35] introduce segmentation tokens, enabling LLMs to output segmentation masks and thus handle pixel-level understanding and reasoning tasks. Nonetheless, they require a large segmentation module, such as SAM [47], making the system highly redundant. As shown in Fig. 2 (d), GLAMM [87] combines the above pipelines to handle object-level and pixel-level tasks. However, this significantly increases the system's *complexity* and *redundancy*. Additionally, GLAMM relies on explicit instructions from the user, **losing** the perception ability to handle basic pixel-level understanding tasks such as instance segmentation, semantic segmentation, panoptic segmentation, and interactive segmentation.

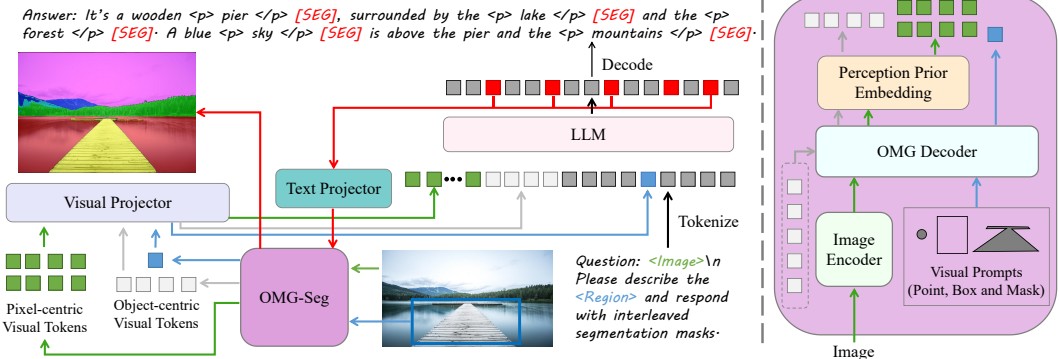

Figure 3: The Overview of OMG-LLaVA. OMG-LLaVA consists of OMG-Seg and LLM. OMG-Seg tokenizes the image into pixel-centric visual tokens, the detected objects, and inputs visual prompts into object-centric visual tokens. Additionally, the [SEG] token output by LLM is decoded by OMG-Seg into segmentation masks. OMG-Seg remains frozen at all stages.

In this paper, we focus on addressing all the challenges above in a more simple yet elegant way. Our OMG-LLaVA unifies image-level (such as image caption and image-based conversation), object-level (such as region caption and visual prompt-based conversation), and pixel-level (such as universal segmentation, referring segmentation, reasoning segmentation, and grounded conversation generation) visual understanding and reasoning tasks into token-to-token generation. The framework follows a simple and elegant system design, including only one visual perception module and one large language model.

**Unified View of Different Tasks.** We model various tasks as the token-to-token generation to bridge the gap between image-level, object-level, and pixel-level understanding and reasoning. To support these tasks, we define three types of tokens: text tokens $T_t$, pixel-centric visual tokens $T_{pv}$, and object-centric visual tokens $T_{ov}$. Text tokens encode textual information. Pixel-centric visual tokens represent dense image features, providing the LLM with comprehensive image information. Object-centric visual tokens encode the features of specified objects, offering the LLM object-centric information, and can be easily decoded into segmentation masks.

Then, all the tasks can be unified as:

$$T_t^{out}, T_{ov}^{out} = LLM(T_{pv}^{in}, T_{ov}^{in}, T_t^{in}) \tag{1}$$

For example, in the classic image-level understanding task, i.e., image caption, a text response $T_t^{out}$ is generated based on text instruction $T_t^{in}$ and image features $T_{pv}^{in}$. In the object-level understanding task, region captioning, the text response $T_t^{out}$ is generated based on text instruction $T_t^{in}$, image features $T_{pv}^{in}$, and specified object-centric visual tokens $T_{ov}^{in}$. The pixel-level reasoning task, referring segmentation, involves generating object-centric visual tokens $T_{ov}^{out}$ based on text instruction $T_t^{in}$ and image features $T_{pv}^{in}$. Additionally, OMG-LLaVA can support various mixed-level tasks, such as providing grounded descriptions around specified objects.

Pixel-centric visual tokens can be obtained by tokenizing images using a CLIP backbone as the tokenizer. However, object-centric visual tokens require encoding object information to be easily decoded into segmentation masks. Therefore, methods like mask pooling in Osprey [126] and ROI pooling in GLaMM [87] fail to meet these requirements. We found that a universal perception decoder can meet all the requirements. Thus, we chose the OMG-Seg decoder [58] as the object-centric tokenizer due to its comprehensive capabilities.

## 3.2 OMG-LLaVA Framework

The framework of OMG-LLaVA is shown in Fig. 2 (e). OMG-LLaVA comprises a large language model (LLM) and a *frozen* universal perception module. The universal perception module encodes images and visual prompts from users into pixel-centric and object-centric visual tokens. It obtains object-centric visual tokens output by the LLM into explicit segmentation mask responses. The LLM accepts text instruction tokens and pixel-centric and object-centric visual tokens from the universal perception module as inputs and then outputs text responses along with object-centric visual tokens. The detailed architecture of OMG-LLaVA is illustrated in Fig. 3. The universal perception module

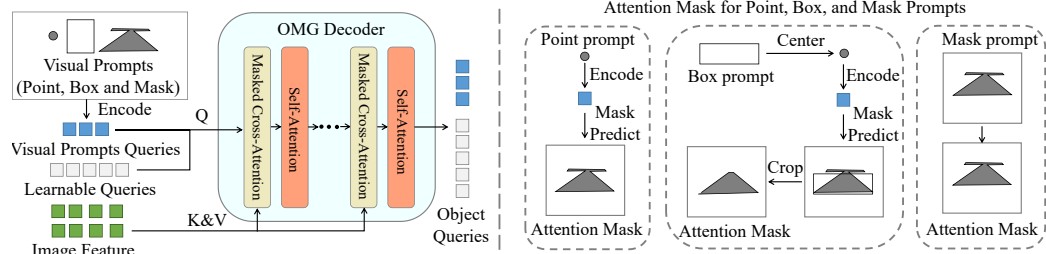

Figure 4: The Architecture of the OMG Decoder. A simple attention mask generation strategy enables the OMG decoder to encode point, box, and mask prompts.

comprises an image encoder, an OMG decoder [58], and a non-trainable perception prior embedding component.

**Image Encoder.** To maximize the perception capabilities of the universal perception module, we use the ConvNeXt-L [75]-based CLIP [86] model as the image encoder and employ a high image resolution (1024×1024). However, the large image resolution results in excessive visual tokens input into the LLM, leading to significantly higher computational costs than using lower-resolution images (such as 224×224 or 336×336). We address this issue by utilizing the lowest resolution image features (32× downsampling). Additionally, we use the pixel shuffle operator to further reduce the image features' resolution. Ultimately, the downsampling factor for the image features used to generate visual tokens is 64, meaning that a 1024×1024 image produces 256 visual tokens.

**OMG Decoder.** We utilize the OMG decoder [58] to generate object-centric visual tokens, furnishing the LLM with information regarding the primary objects in the image and those mentioned by the user's input visual prompts. As shown on the left side of Fig. 4, the OMG decoder comprises masked cross-attention [19] and self-attention layers. The OMG decoder's input includes a set of learnable object queries [20; 19; 10] for automatically capturing all objects of interest and visual prompt queries derived from encoded input visual prompts [47]. The visual prompt queries and learnable object queries are collectively termed object queries. The OMG decoder probes feature for object queries from the image features by employing masked cross-attention and models relationships between objects through self-attention. The object queries can be decoded into segmentation masks and object categories via a simple FFN layer. With the OMG decoder, OMG-LLaVA can efficiently tokenize object information into object-centric visual tokens, thereby equipping the LLM with information about objects in the image and those referenced by the user.

The OMG decoder can accept point prompts as input. While box and mask prompts can be easily converted into point prompts, this crude conversion significantly loses prompt information, complicating the explicit encoding of the user's intent. To address this, we can impose constraints on the attention masks of the masked cross-attention layers based on the visual prompt to precisely encode the object information referenced by the prompt. As depicted on the right side of Fig. 4, we utilize the box coordinates to define attention masks for all pixel features outside the box for box prompts. Similarly, we directly employ the provided object mask to generate attention masks for mask prompts. With this straightforward attention mask modification strategy, OMG-LLaVA can accurately capture the user's visual prompts, encompassing point, box, and mask prompts.

**Perception Prior Embedding.** We find that directly combining a frozen perception module with LLM doesn't perform well, as also observed in LISA [49]. To retain the full capabilities of the universal perception module, OMG-LLaVA doesn't fine-tune the perception module to adapt to the output of the large language model. Instead, we propose

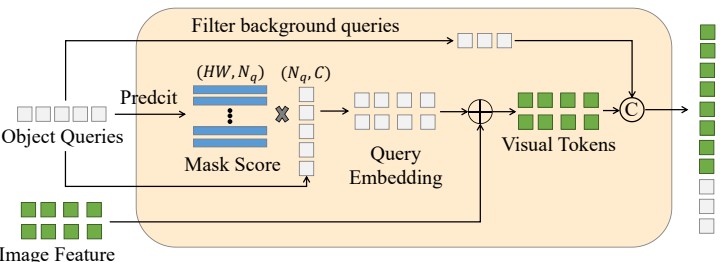

Figure 5: The process of the perception prior embedding strategy. The perception prior embedding strategy integrates object queries into image features based on segmentation prior.

a perception prior embed-
ding strategy to tackle this challenge. Fig. 5 illustrates the perception prior embedding strategy.

First, we fuse the image features $\mathcal{F} \in \mathbb{R}^{HW \times C}$ outputted by the image encoder with the object queries $\mathcal{Q} \in \mathbb{R}^{N_q \times C}$ outputted by the OMG decoder $\mathcal{D}$. Specifically, we utilize the segmentation mask $\mathcal{M} \in \mathbb{R}^{N_q \times HW}$ obtained from the object queries and the corresponding confidence score $\mathcal{S} \in \mathbb{R}^{1 \times N_q}$ to derive a mask score $MS \in \mathbb{R}^{HW \times N_q}$ for each pixel for the object queries:

$$MS = Softmax(\mathcal{M} \odot \mathcal{S}, dim = -1) \tag{2}$$

Then, we compute a weighted average of the object queries $\mathcal{Q}$ based on the mask score $MS$ and obtain the corresponding weighted object queries for each pixel. Pixel-centric visual tokens $T_{pv}$ are obtained by adding the weighted object queries to the image features $\mathcal{F}$:

$$T_{pv} = MS \cdot \mathcal{Q} + \mathcal{F} \tag{3}$$

Additionally, we treat the foreground object queries as object-centric visual tokens $T_{ov}$. The object-centric visual tokens $T_{ov}$ are concatenated with the pixel-centric visual tokens $T_{pv}$ to form the visual tokens $T_v = (T_{pv}, T_{ov})$, which are input to the LLM to provide rich perception prior information.

**Visual Projector and Text Projector.** Following [71], we use an MLP as the visual projector, which is responsible for mapping visual tokens to the LLM's text embedding space. Since our visual tokens are pixel-centric and object-centric tokens, the visual projector comprises two MLPs, each handling one type of visual token separately. Inspired by [49; 87], we also use a simple MLP to map the LLM output's hidden states of the [SEG] token to the visual space.

**Instruction Formulation.** OMG-LLaVA can accept **visual** input, **text** input, and **visual prompt** input and output text responses and segmentation token, segmentation masks and labels. Thus, it can handle tasks such as image captioning, image-based conversation, region captioning, visual prompt-based conversation, referring segmentation, reasoning segmentation, grounded conversation, etc. We use a unified instruction formulation to support these functionalities. As shown in Fig. 3, there are three special tokens: <Image>, <Region>, and [SEG]. Before being fed into the LLM, the <Image> token is replaced by visual tokens $T_v$, and the <Region> token can be replaced by any object-centric visual token encoded by the visual prompt. The [SEG] token in the LLM's output is sent to the frozen OMG decoder to be decoded into a segmentation mask.

### 3.3 Training and Testing Setup

**Training.** Following LLaVA [71], our OMG-LLaVA performs two-stage training: pretraining and instruction tuning. During the pretraining stage, the perception model and LLM are frozen, and only the visual and text projectors can be tuned. In addition to the text regression loss, we apply regularization penalties to the visual projector $\mathcal{P}_v$ and text projector $\mathcal{P}_t$ to preserve object-centric information as much as possible.

$$\mathcal{L}_{pretrain} = \mathcal{L}_{text} + \mathcal{L}_{reg}, \quad \mathcal{L}_{reg} = (T_{ov} - \mathcal{P}_t(\mathcal{P}_v(T_{ov})))^2 \tag{4}$$

During instruction tuning, in addition to finetuning the visual projector and text projector, we use LoRA [37] to finetune the LLM. Following [87; 58], besides the text regression loss, we apply cross-entropy loss and dice loss [80] to supervise the segmentation mask decoded by the [SEG] token, as shown in following (We set $\alpha = 5$ $\beta = 2$ by default):

$$\mathcal{L}_{intruction} = \mathcal{L}_{text} + \mathcal{L}_{mask}, \quad \mathcal{L}_{mask} = \alpha \mathcal{L}_{CE} + \beta \mathcal{L}_{DICE} \tag{5}$$

**Testing.** The image-level, object-level, and pixel-level understanding and reasoning tasks can all be encompassed within the Eq. 3.1 paradigm. During the inference stage, we encode the necessary task requirements, such as text prompts, visual prompts, and image features, into tokens to input into the LLM. The output tokens of LLM are then decoded into text responses and segmentation mask responses according to the task definition. We refer the readers to check the more details in the appendix.

Table 2: The comprehensive comparison of OMG-LLaVA and other MLLMs regarding pixel-level and object-level understanding and reasoning capability and performance. "-" indicates that the method does not handle this task. † indicates that the method used the GranD dataset [87] for pretraining, which is significantly larger than the datasets used by other methods.

| Method | Visual Encoder Num | COCO PQ | VIPseg VPQ | refCOCO cIoU | refCOCO+ cIoU | GCG METEOR | AP50 | refCOCOg(C) METEOR |
|---|---|---|---|---|---|---|---|---|
| OSprey [126] | 1 | - | - | - | - | - | - | 16.6 |
| LISA [49] | 2 | - | - | 74.1 | 62.4 | 13.0 | 25.2 | - |
| NeXT-Chat [128] | 2 | - | - | 74.7 | 65.1 | - | - | 12.0 |
| LaSagnA [102] | 2 | - | - | 76.8 | 66.4 | - | - | - |
| GSVA [109] | 2 | - | - | 76.4 | 64.5 | - | - | - |
| AnyRef [35] | 2 | - | - | 74.1 | 64.1 | - | - | 16.2 |
| GLaMM† [87] | 2 | - | - | 79.5 | 72.6 | 15.2 | 28.9 | 15.7 |
| PixelLM [88] | 1 | - | - | 73.0 | 66.3 | - | - | - |
| OMG-LLaVA | 1 | 53.8 | 49.8 | 78.0 | 69.1 | 14.9 | 29.9 | 15.3 |

Table 3: Performance on referring expression segmentation datasets. The evaluation metric is cIoU. "ft" indicates finetuning on the referring expression datasets.

| Method | Freeze Decoder | Visual Encoder | refCOCO Val | TestA | TestB | refCOCO+ Val | TestA | TestB | refCOCOg Val | Test |
|---|---|---|---|---|---|---|---|---|---|---|
| LISA [49] | × | 2 | 74.1 | 76.5 | 71.1 | 62.4 | 67.4 | 56.5 | 66.4 | 68.5 |
| LISA(ft) [49] | × | 2 | 74.9 | 79.1 | 72.3 | 65.1 | 70.8 | 58.1 | 67.9 | 70.6 |
| PixelLM [88] | × | 1 | 73.0 | 76.5 | 68.2 | 66.3 | 71.7 | 58.3 | 69.3 | 70.5 |
| GSVA(ft) [109] | × | 2 | 77.2 | 78.9 | 73.5 | 65.9 | 69.6 | 59.8 | 72.7 | 73.3 |
| OMG-LLaVA | ✓ | 1 | 75.6 | 77.7 | 71.2 | 65.6 | 69.7 | 58.9 | 70.7 | 70.2 |
| OMG-LLaVA(ft) | × | 1 | 78.0 | 80.3 | 74.1 | 69.1 | 73.1 | 63.0 | 72.9 | 72.9 |
| OMG-LLaVA(ft) | ✓ | 1 | 77.2 | 79.8 | 74.1 | 68.7 | 73.0 | 61.6 | 71.7 | 71.9 |

Table 4: Performance on grounded conversation generation datasets. "ft" indicates finetuning on the GranDf [87] dataset. † indicates that the method used the GranD dataset [87] for pretraining.

| Methods | ft | Visual Encoder | Val METEOR | CIDEr | AP50 | mIOU | Test METEOR | CIDEr | AP50 | mIOU |
|---|---|---|---|---|---|---|---|---|---|---|
| Kosmos-2 [82] | ✓ | 1 | 16.1 | 27.6 | 17.1 | 55.6 | 15.8 | 27.2 | 17.2 | 56.8 |
| LISA [49] | ✓ | 2 | 13.0 | 33.9 | 25.2 | 62.0 | 12.9 | 32.2 | 24.8 | 61.7 |
| GLaMM† [87] | ✓ | 2 | 15.2 | 43.1 | 28.9 | 65.8 | 14.6 | 37.9 | 27.2 | 64.6 |
| OMG-LLaVA | × | 1 | 13.8 | 36.2 | 26.9 | 64.6 | 13.5 | 33.1 | 26.1 | 62.8 |
| OMG-LLaVA | ✓ | 1 | 14.9 | 41.2 | 29.9 | 65.5 | 14.5 | 38.5 | 28.6 | 64.7 |

Table 5: Ablation study on RES and GCG datasets.

| Methods | refCOCO cIoU | gIoU | refCOCO+ cIoU | gIoU | refCOCOg cIoU | gIoU | GCG METEOR | mIoU |
|---|---|---|---|---|---|---|---|---|
| Baseline (M0) | 58.7 | 61.0 | 52.6 | 55.0 | 55.8 | 58.1 | 13.2 | 51.0 |
| + Perception prior embedding (M1) | 72.5 | 74.3 | 63.2 | 65.4 | 67.8 | 70.6 | 13.6 | 62.1 |
| + Object query input (M2) | 74.4 | 75.9 | 64.4 | 66.2 | 68.5 | 71.5 | 13.8 | 63.6 |

# 4 Experiment

**Dataset Setup.** During the pretraining stage, we use the LLaVA pretraining dataset [71] to perform visual-text alignment, following LLaVA. The instruction tuning process of OMG-LLaVA involves a diverse range of tasks and datasets. For image-level understanding and reasoning tasks, we use the LLaVA dataset [71; 72; 70], which includes 665K descriptions, reasoning, and conversation data. For object-level understanding and reasoning, we use the object-level description and conversation data from the Osprey dataset [126] and the object-level point-prompt data from the MDVP dataset [67], which contain approximately 74K and 200K data, respectively. For pixel-level understanding and reasoning, we use the referring segmentation datasets, including refCOCO, refCOCO+ [42], refCOCOg [121], and refClef, totaling 74K data. Additionally, semantic segmentation datasets, including ADE20k [137] and COCO-stuff [7], totaling 26K data, and the grounded conversation generation dataset GranDf [87], containing 200K data, are used.

**Implementation Details.** We use the pre-trained ConvNext-L [75] OMG-Seg [58] as the universal perception module and InterLM2-7B [9] as the LLM for OMG-LLaVA. We adopt xtuner codebase [22] to build our model and data pipeline. The image is resized to 1024×1024. During the pretraining stage, only the visual projector and text projector are trained, with an initial learning rate set to 1e-3. During the instruction tuning stage, the initial learning rate is set to 2e-4, with only the perception model kept frozen, and the LLM is fine-tuned using LoRA [37]. The maximum sequence length in the LLM is set to 2,048. All training is conducted on four NVIDIA A800 GPUs with 80GB of memory. The pretraining stage and instruction tuning stage took 7 hours and 48 hours, respectively.

## 4.1 Main Results

**Comparison with MLLMs.** OMG-LLaVA is comprehensively compared with current MLLMs with perception capabilities, and the results are shown in Tab. 2. OMG-LLaVA demonstrates the most comprehensive capabilities. It achieves performance comparable to the SOTA in referring segmentation, grounded conversation generation, and region captioning. Additionally, OMG-LLaVA retains basic segmentation ability, enabling it to handle universal image and video segmentation tasks. Compared to other MLLMs, OMG-LLaVA features a simple and elegant system design, incorporating only a single visual encoder.

**Referring Expression Segmentation.** We evaluate OMG-LLaVA on refCOCO, refCOCO+, and refCOCOg, with the results shown in Tab. 3. OMG-LLaVA outperforms LISA [49] by 1.5 cIoU, 3.2 cIoU, and 4.3 cIoU on the validation sets of refCOCO, refCOCO+, and refCOCOg, respectively, while keeping the OMG decoder frozen and using only a single visual encoder. When we unfreeze the OMG decoder and finetune OMG-LLaVA on the referring expression segmentation task, OMG-LLaVA achieves 78.0, 69.1, and 72.9 cIoU on refCOCO, refCOCO+, and refCOCOg, respectively, surpassing LISA by 3.1, 4.0, and 5.0 cIoU. Compared to PixelLM [88], OMG-LLaVA shows performance improvements of 5.0 cIoU and 3.6 cIoU on refCOCO and refCOCOg, respectively.

**Grounded Conversation Generation.** Grounded conversation generation is a comprehensive and complex task that involves both image-level and pixel-level understanding and reasoning. MLLMs need to have the ability to provide fine-grained image descriptions and pixel-level understanding, linking the objects in the image captions to the corresponding segmentation masks. As shown in Tab. 4, when trained with comparable data, OMG-LLaVA surpasses LISA [49] by 1.9 METEOR and 7.3 CIDEr in image description ability. In terms of pixel understanding, OMG-LLaVA also outperforms LISA by 4.7 AP50 and 3.5 mIoU, even though LISA uses SAM and finetunes its segmentation decoder. Despite GLaMM [87] using much more training data than OMG-LLaVA, OMG-LLaVA demonstrates comparable pixel-understanding capabilities, outperforming GLaMM with 0.6 CIDEr, 1.4 AP50 and 0.1 mIoU on the test set.

## 4.2 Ablation and Analysis

**Ablation Study.** We conduct ablation studies on referring expression segmentation and grounded conversation generation datasets, with all training and testing settings consistent with the main experiments. We use a simple combination of OMG-Seg [58] and LLaVA [71] as our baseline, similar to LISA [49], where the [SEG] tokens output by the LLM were input into OMG-Seg to obtain segmentation masks, with OMG-Seg kept frozen.

As shown in Tab. 5, the baseline performed poorly on the RES datasets. Similarly, it exhibited low segmentation quality on the GCG dataset. This is because the LLM did not acquire any segmentation priors and needed to generate segmentation queries based on image features and adapt them to the input of the frozen perception module, which is a challenging task.

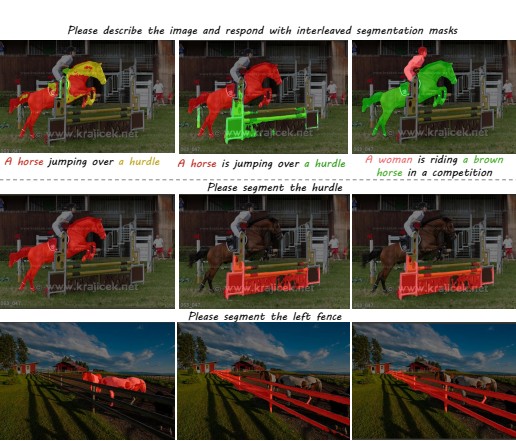

Figure 6: Visualization of the effectiveness of the proposed strategies. The left part shows the baseline (M0 in Tab. 5), the middle part shows the model with perception prior embedding (M1 in Tab. 5), and the right part shows the model with both perception prior embedding and object query input (M2 in Tab. 5).

When using our proposed perception prior embedding strategy, OMG-LLaVA exhibits performance gains of 13.8 cIoU, 10.6 cIoU, and 11.7 cIoU on refCOCO, refCOCO+, and refCOCOg, respectively. Additionally, the perception prior embedding strategy also brings a performance improvement of 11.1 mIoU on the GCG dataset and a slight improvement in image description capability (0.4 METEOR). When foreground object queries were provided to the LLM, OMG-LLaVA further improved its performance by 1.9 cIoU on refCOCO and 1.5 mIoU on GCG.

We conducted a visualization analysis of the proposed strategies. As shown in the left part of Fig. 6, the simple baseline has poor capability in associating text and segmentation, which is the crucial reason for its poor performance on RES. When using our proposed perception prior embedding strategy, the object query and pixel features are explicitly integrated according to the perception prior, resulting in significantly enhanced text-segmentation association capability. By adopting the object query input strategy, the quality of some challenging segmentation cases, such as the lower right corner of the fence in Fig 6, slightly improves.

**Qualitative Results.** We provide visualization results of OMG-LLaVA on multiple image-level, object-level, and pixel-level tasks in Fig. 1. Additional qualitative visualization results or comparable visual results for referring expression segmentation and grounded conversation generation are presented in the appendix.

## 5 Conclusion

We present a new MLLM, OMG-LLaVA, which bridges image-level, object-level, and pixel-level understanding and reasoning in one model. Our method only contains one image encoder, one LLM, and one decoder. With proposed perception prior embedding and unified task instruction tuning, OMG-LLaVA can perform over 8 different multi-modal learning tasks, as well as preserving the visual perception ability of the OMG-Seg baseline. Our method can achieve comparable results compared with previous combined works with much fewer trainable parameters and computation costs. We hope our work can inspire the community to rethink the design of the MLLM meta-architecture to minimize the model components and maximize the MLLM's functionalities.

**Acknowledgment.** This work was supported by the National Natural Science Foundation of China (grant No. 42171430). This work is also supported by the CCF-Kuaishou Large Model Explorer Fund. It is also partially supported under the RIE2020 Industry Alignment Fund Industry Collaboration Projects (IAF-ICP) Funding Initiative, as well as cash and in-kind contributions from the industry partner(s).

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

Table 6: Performance on image-level benchmarks.

| Method | MME [32] | MMBench [74] | SEED-Bench [50] | POPE [63] | AI2D [43] |
|---|---|---|---|---|---|
| Training only with LLaVA dataset | | | | | |
| LLaVA 1.5 [69] | 1422/267 | 68.5 | 65.9 | 86.7 | 56.6 |
| OMG-LLaVA | 1448/282 | 67.5 | 68.9 | 89.7 | 61.7 |
| Co-training with LLaVA dataset and segmentation datasets | | | | | |
| LISA [49] | 1/1 | 0.4 | - | 0.0 | 0.0 |
| PixelLM [88] | 309/135 | 17.4 | - | 0.0 | 0.0 |
| LaSagnA [102] | 0/0 | 0.0 | - | 0.0 | 0.0 |
| GLaMM [87] | 14/9 | 36.8 | - | 0.94 | 28.2 |
| OMG-LLaVA | 1177/235 | 47.9 | 56.5 | 80.0 | 42.9 |

Table 7: Performance with different LLMs.

| LLM | refCOCO | | refCOCO+ | | MME | | MMBench | SEED-Bench | POPE | AI2D | MMstar | SQA |
|---|---|---|---|---|---|---|---|---|---|---|---|---|
| | CIoU | GIoU | CIoU | GIoU | perception | reasoning | | | | | | |
| Phi3-3.8B [1] | 76.5 | 78.0 | 67.8 | 70.0 | 1291.6 | 265.0 | 59.6 | 60.6 | 86.7 | 56.9 | 37.1 | 64.7 |
| InternLM2-7B [9] | 76.3 | 77.8 | 67.7 | 69.9 | 1177.1 | 235.4 | 47.9 | 56.5 | 80.0 | 42.9 | 33.1 | 57.8 |
| Qwen2-7B [113] | 76.7 | 78.2 | 69.1 | 71.2 | 1215.7 | 251.1 | 62.8 | 60.7 | 84.3 | 52.6 | 37.2 | 66.4 |

# A    Appendix

**Overview.** In this appendix, we will first give more implementation and training details of our method. Then, we present more detailed ablation studies on several component designs. Next, we present more detailed visualization results. In the end, we discuss the limitations and future work.

## A.1    More Implementation Details

**Pre-training.** Following LLaVA, OMG-LLaVA first performs pre-training to learn the projector that projects visual tokens into the text space. During the pre-training stage, we freeze the visual encoder, OMG head, and LLM to train the visual projector for projecting visual tokens into the text space and to train the text projector for restoring the projected object-centric visual tokens to the segmentation embedding. The training data used in the pre-training stage is the same as that used in LLaVA. In this stage, the OMG-LLaVA is trained for 1 epoch. The batch size is 256, with 32 per GPU, and the learning rate is 0.001.

**Supervised fine-tuning.** During the instruction tuning stage, we freeze the visual encoder and OMG head, finetune the LLM using LoRA, and fully finetune the text and visual projectors. We train OMG-LLaVA for 1 epoch on all instruction tuning datasets, including the LLaVA instruction tuning dataset, referring expression segmentation datasets, semantic segmentation datasets, grounded conversation generation datasets, mask-based visual prompt datasets, and point-based visual prompt datasets. The batch size is 128, with 16 per GPU, and the learning rate is 2e-4.

**Inference details for each task.** OMG-LLaVA generates answers token by token during the inference stage based on the given question. We use a fixed template for the referring expression segmentation task to create the question: "*Please segment {EXPRESSION} in this image.*" In rare cases where OMG-LLaVA does not predict the [SEG] token, we use an empty mask as the segmentation result. We use the fixed question for the grounded conversation generation task: "*Could you please give me a detailed description of the image? Please respond with interleaved segmentation masks for the corresponding parts of the answer.*" For other tasks, we remove special tokens such as <p>, </p>, and [SEG] from OMG-LLaVA's responses to ensure the answers contain only text.

## A.2    More Experiment Results.

**Evaluation results on image-level benchmarks.** We evaluate OMG-LLaVA on several image-level benchmarks, including MME [32], MMBench [74], SEED-Bench [50], POPE [63], and AI2D [43] benchmarks. The evaluation results are shown in Tab. 6. When jointly Co-training with image-level and pixel-level datasets, OMG-LLaVA achieves 1412, 47.9, 46.5, 80.0 and 42.9 on MME, MMBench, SEED-Bench, POPE and AI2D benchmarks, respectively. Compared with GLaMM [87], PixelLM [88], and LISA [49], OMG-LLaVA demonstrates significant performance improvement.

Table 8: Ablation study of projector for object-centric visual tokens.

| Methods | Cross Attn. | Individual | refCOCO | | refCOCO+ | | refCOCOg | | refCOCOg(C) |
|---|---|---|---|---|---|---|---|---|---|
| | | | cIoU | gIoU | cIoU | gIoU | cIoU | gIoU | METEOR |
| Baseline (M0) | | | 74.5 | 75.9 | 63.6 | 65.9 | 68.7 | 71.0 | 13.6 |
| M1 | ✓ | | 72.3 | 73.7 | 60.6 | 63.0 | 66.5 | 69.2 | 13.2 |
| M2 | | ✓ | 72.3 | 74.1 | 60.8 | 63.5 | 65.4 | 68.6 | 13.1 |

Table 9: Ablation study on answer format of segmentation-based tasks. The first row represents the RES task using the fixed answer: "*Sure, it is [SEG]*." and the GCG task using "*<p> Expression </p> [SEG]*." The second row represents the segmentation tasks' answer format being unified as "*<p> Expression </p> [SEG]*.

| Format | refCOCO | | refCOCO+ | | refCOCOg | | GCG | | |
|---|---|---|---|---|---|---|---|---|---|
| | cIoU | gIoU | cIoU | gIoU | cIoU | gIoU | METEOR | AP50 | mIOU |
| It is [SEG]. | 75.5 | 76.5 | 65.8 | 67.8 | 70.6 | 72.3 | 13.8 | 27.3 | 64.4 |
| <p> Expression </p> [SEG]. | 75.6 | 76.8 | 65.6 | 67.6 | 70.7 | 72.6 | 13.8 | 26.9 | 64.6 |

Table 10: Ablation study on segmentation embeddings.

| | refCOCO | | refCOCO+ | | refCOCOg | |
|---|---|---|---|---|---|---|
| | cIoU | gIoU | cIoU | gIoU | cIoU | gIoU |
| Last layer's hidden state | 74.3 | 75.5 | 64.5 | 66.4 | 70.0 | 71.9 |
| Mean of all layers' hidden states | 74.3 | 75.8 | 64.5 | 66.4 | 68.7 | 71.4 |
| Concatenate of all layers' hidden states | 70.0 | 71.1 | 61.8 | 63.6 | 62.3 | 64.4 |

When training only with the LLaVA [69] dataset, OMG-LLaVA achieves 1730, 67.5, 68.9, 89.7, and 61.7 on MME, MMBench, SEED-Bench, POPE, and AI2D benchmarks. OMG-LLaVA outperforms LLaVA-1.5 [69] with 41, 3.0, 3.0, 5.1 on MME, SEED-Bench, POPE, and AI2D benchmarks with the same training data.

**Performance with diverse LLMs.** We construct the OMG-LLaVA using diverse LLMs. The performance is shown in Tab. 7. In addition to InternLM2 [9], we have tried using PHI-3.8b [1] and Qwen2-7B [113], which achieved better performance on pixel-level and image-level benchmarks than InternLM2. When using the stronger Qwen2-7B, OMG-LLaVA achieves 76.7 cIoU and 69.1 cIoU on RefCOCO and RefCOCO+ benchmarks, and 1466.8, 62.8, 60.7, 84.3, 52.6, 37.2, and 66.4 on MME [32], MMBench [74], SEED-Bench [50], POPE [63], AI2D [43], MMstar [15] and SQA [76] benchmarks.

### A.3 More Detailed Ablation Studies.

**Projector for object-centric visual tokens.** We conducted ablation experiments on the vision projector. The results are shown in Tab. 8. We use a simple MLP projector as the baseline for object-centric visual tokens. When we added a cross-attention layer to the projector, performance on segmentation and visual prompt-based tasks decreased. This is because the introduction of the cross-attention layer caused the object-centric visual tokens to incorporate too many pixel-centric visual tokens, leading to interference with the object information. Furthermore, when the projector for object-centric visual tokens generated from visual prompt input and object queries is not shared, performance declines on segmentation and visual prompt-based tasks. Therefore, a shared MLP projector can effectively project object-centric visual tokens into the text space.

**Answer format for segmentation-based tasks.** In LISA [49], the response for the referring expression segmentation task is fixed as "*Sure, it is [SEG]*." However, this fixed answer may interfere with the instruction-following ability of the LLM, leading it to respond with "*Sure, it is [SEG]*." for new instructions. In GLaMM [87], for the grounded conversation generation task, the response is typically "*<p> Expression </p> [SEG]*." Since the "*Expression*" is flexible and variable, the LLM is less likely to overfit to a fixed response.

We conduct ablation experiments on the answer format for segmentation tasks, and the results are shown in Tab. 9. We find that unifying the answer format for segmentation tasks (including RES and GCG) as "*<p> Expression </p> [SEG]*" yields better performance. This more flexible answer

Express: *the grass*   *the sand*   *the table*   *the smallest chair*
*beside the black sofa*

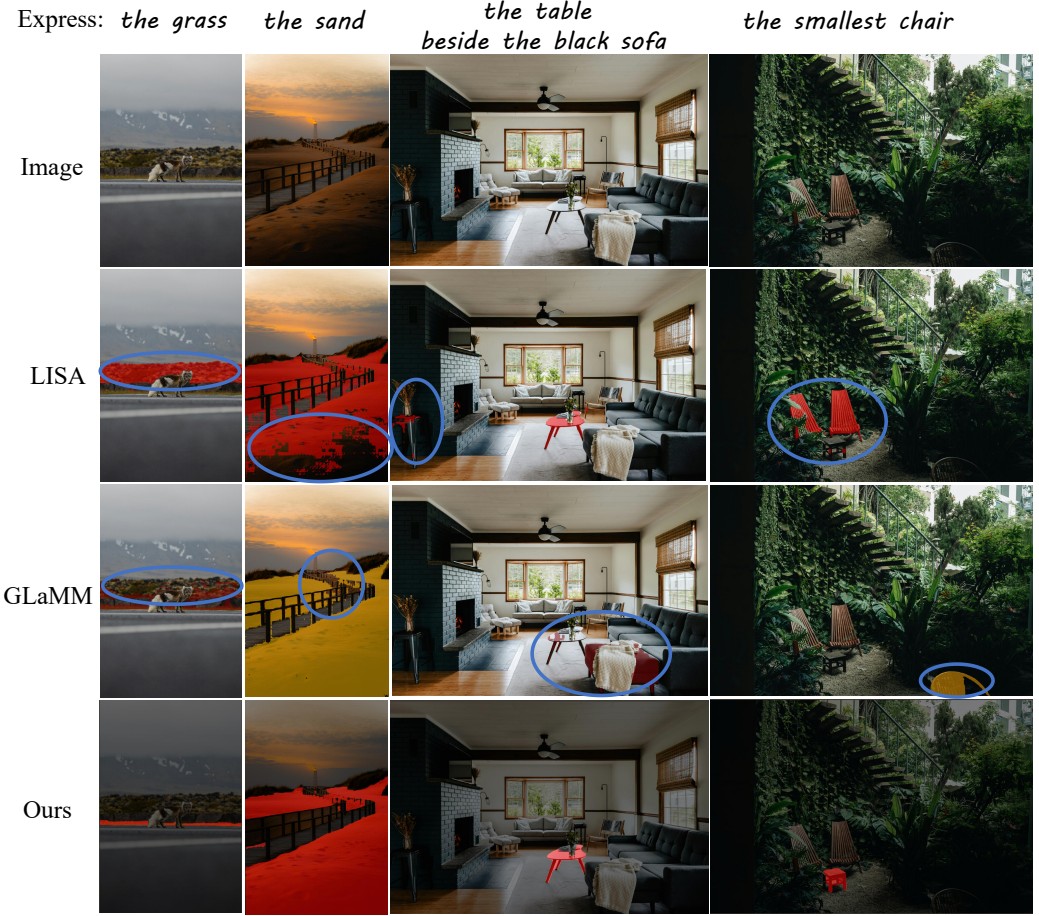

Figure 7: Qualitative comparison on the referring expression segmentation task. LISA uses the 13B LLM, while GLaMM and our proposed OMG-LLaVA use the 7B LLM.

format not only achieves better performance in the referring expression segmentation task compared to the fixed answer but also avoids the damage to the LLM's instruction-following ability.

**Segmentation embeddings.** We conduct ablation experiments on the generation strategy of segmentation embedding, and the results are shown in Tab. 10. We explore whether the hidden states of the intermediate layers corresponding to the [SEG] token are helpful for segmentation. Compared to using the hidden states of the last layer of the [SEG] token as the segmentation embedding, using the mean of the hidden states from all layers as the segmentation embedding resulted in negligible improvement on refCOCO but led to a significant performance drop on the more challenging refCOCOg. Concatenating the hidden states from all layers of the [SEG] token as the segmentation embedding resulted in a significant performance drop across all RES tasks. Therefore, the hidden state of the last layer already contains sufficient features to generate the segmentation mask, and introducing hidden states from other intermediate layers does not yield better segmentation results.

### A.4   More Visualization Results

**Qualitative comparison with SOTA methods.** We conduct qualitative comparisons and analyses on various tasks, including referring expression segmentation, grounded conversation generation, and image-based conversation, against the SOTA methods LISA [49] and GLaMM [87]. Fig. 7 shows the visualization results of the RES task for LISA, GLaMM, and our proposed OMG-LLaVA. OMG-LLaVA demonstrates a more stable segmentation performance than LISA and GLaMM. Additionally, OMG-LLaVA exhibits better image and text understanding capabilities than LISA (13B) and GLaMM, as illustrated in the fourth column with the example of "*the smallest chair*".

Fig. 8 shows the visualization results of the GCG task for GLaMM [87] and OMG-LLaVA. Our proposed OMG-LLaVA provides more detailed and accurate descriptions of the scene, such as "*lighthouse*" and "*bear*." Additionally, OMG-LLaVA demonstrates more stable segmentation capabilities, as seen in the "*mountain*" in the bottom-right corner image.

Fig. 9 shows the visualization results of the visual prompt-based description task for GLaMM [87] and OMG-LLaVA. Compared to GLaMM, OMG-LLaVA supports more flexible visual prompts, including point, box, and mask prompts. Additionally, OMG-LLaVA can generate more detailed object captions and demonstrate a more accurate image understanding.

Fig. 10 shows the visualization results of the image-based conversation task for LISA [49], GLaMM [87], and OMG-LLaVA. Compared to LISA and GLaMM, OMG-LLaVA has stronger instruction-following ability. For example, when answering the question, "*What is the number on the jersey of the athlete squatting on the ground?*" both LISA and GLaMM incorrectly segmented "*the jersey of the athlete squatting on the ground*." Compared to GLaMM, OMG-LLaVA can provide more detailed and accurate answers to user questions. Compared to LISA, OMG-LLaVA demonstrates stronger scene understanding and reasoning abilities. For instance, in question 3 of Fig. 10, LISA gave an utterly incorrect answer despite using a larger LLM (13B).

**Visualization results of RES.** We provide additional visualization results of OMG-LLaVA on the RES task in Fig. 11. OMG-LLaVA demonstrates a strong understanding of spatial relationships and human actions, enabling it to accurately and reliably segment the specified objects based on these descriptions. Furthermore, even without training on any reasoning segmentation data, OMG-LLaVA exhibits the ability to perform reasoning segmentation. As shown in Fig. 12, OMG-LLaVA can infer the target based on the question and accurately segment the corresponding object.

**Visualization results of GCG.** As depicted in Fig. 13, our method performs well on the grounded conversation generation task. OMG-LLaVA demonstrates strong scene understanding and object segmentation capabilities. Although some objects are overlooked, this is due to the omission of many objects in the image captions of the Grandf dataset. We believe that using higher-quality data for training would result in even better performance for OMG-LLaVA.

**Visualization results of visual prompts-based description.** Fig. 14 shows more visualization results for the visual prompt-based description task. OMG-LLaVA supports input of point, box, and mask-based visual prompts and provides detailed descriptions. These descriptions include information about the objects and their relationships with other objects in the scene.

## A.5 Limitation and Future Work Discussion

**Limitations of OMG-LLaVA.** Although OMG-LLaVA achieves image-level, object-level, and pixel-level capabilities with a concise and elegant architecture, much room still exists for improvement. Firstly, joint training with pixel-level understanding data often leads to decreased image-level capability, a phenomenon widely observed in LISA [49] and GLaMM [87]. This challenge could be addressed by organizing the data to eliminate this conflict. Secondly, due to the lack of multi-granularity segmentation capability in OMG-Seg, OMG-LLaVA cannot perform part-level segmentation. This challenge could be addressed using a more powerful and universal perception module by adding part-level visual inputs.

**Future Works.** Several future directions can be explored with our new meta-architecture. We list two potential directions, including video and more instruction-tuning data. Although OMG-Seg [58] can acquire the video inputs, OMG-LLaVA still cannot perform pixel-level spatial-temporal reasoning. This is due to the lack of such datasets. Moreover, more instruction-tuning data involve more localization outputs, and multiple round conversations can be used to build a stronger MLLM model. For example, we plan to use full GLaMM datasets [87] and more detection datasets [48; 89] for joint co-training as future work if more computation resources are available.

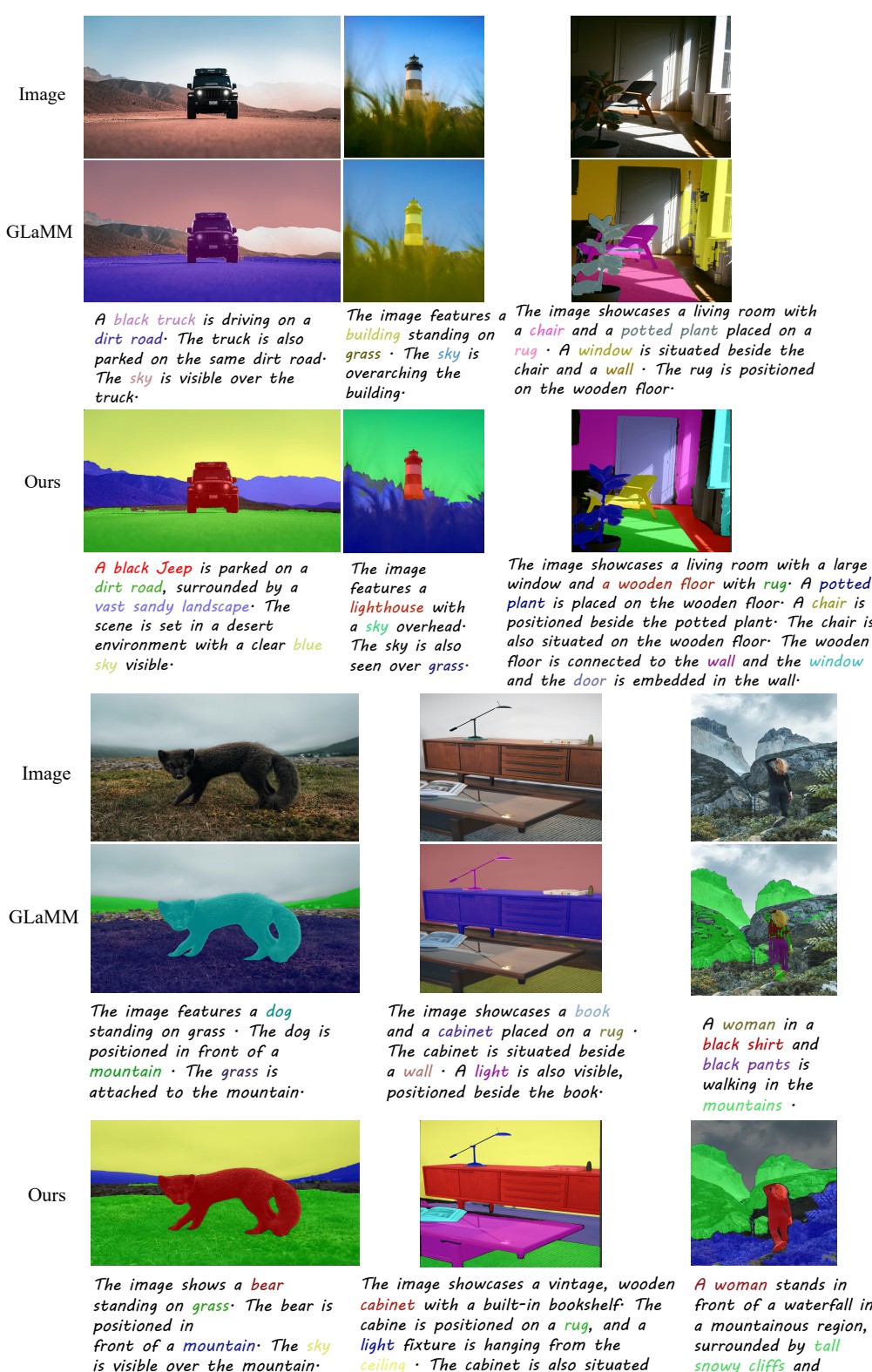

Figure 8: Qualitative comparison on the grounded conversation generation task.

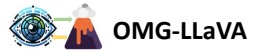 **OMG-LLaVA**   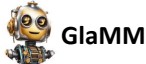 **GlaMM**

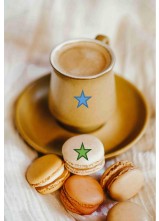

*A coffee cup is prominently positioned in the image. It's a large cup with a brown handle, and it's placed on a saucer. The cup is filled with a warm beverage, likely coffee, and it's the main focus of the scene.*

*A small, round pastry is located in the middle of the plate. It's the second pastry from the left and is the second row from the bottom.*

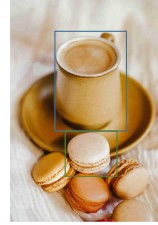

*a cup of coffee*

*a macaroon on the left of the plate*

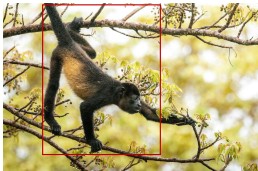

*A small monkey is seen in the image, hanging from a branch. The monkey appears to be a baby, possibly a baby monkey, and is positioned towards the left side of the image.*

*A large, leafy tree is situated in the foreground of the image. It appears to be a mature tree, possibly a pine tree, and is located near a house. The tree is the main focus of this region, and it stands out against the backdrop of the house.*

*A silver car is parked on the side of the road, occupying a significant portion of the image. The car's color is white, and it appears to be a sedan. The car is positioned in front of a blue house, which is also visible in the scene.*

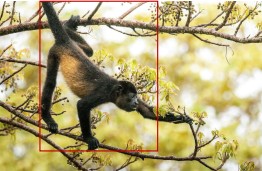

*monkey hanging from tree branch*

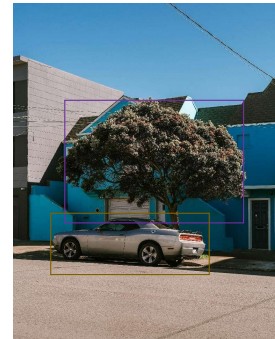

*a tree in front of a house*

*a grey car parked on the side of the road*

Figure 9: Qualitative comparison on the visual prompt-based description task.

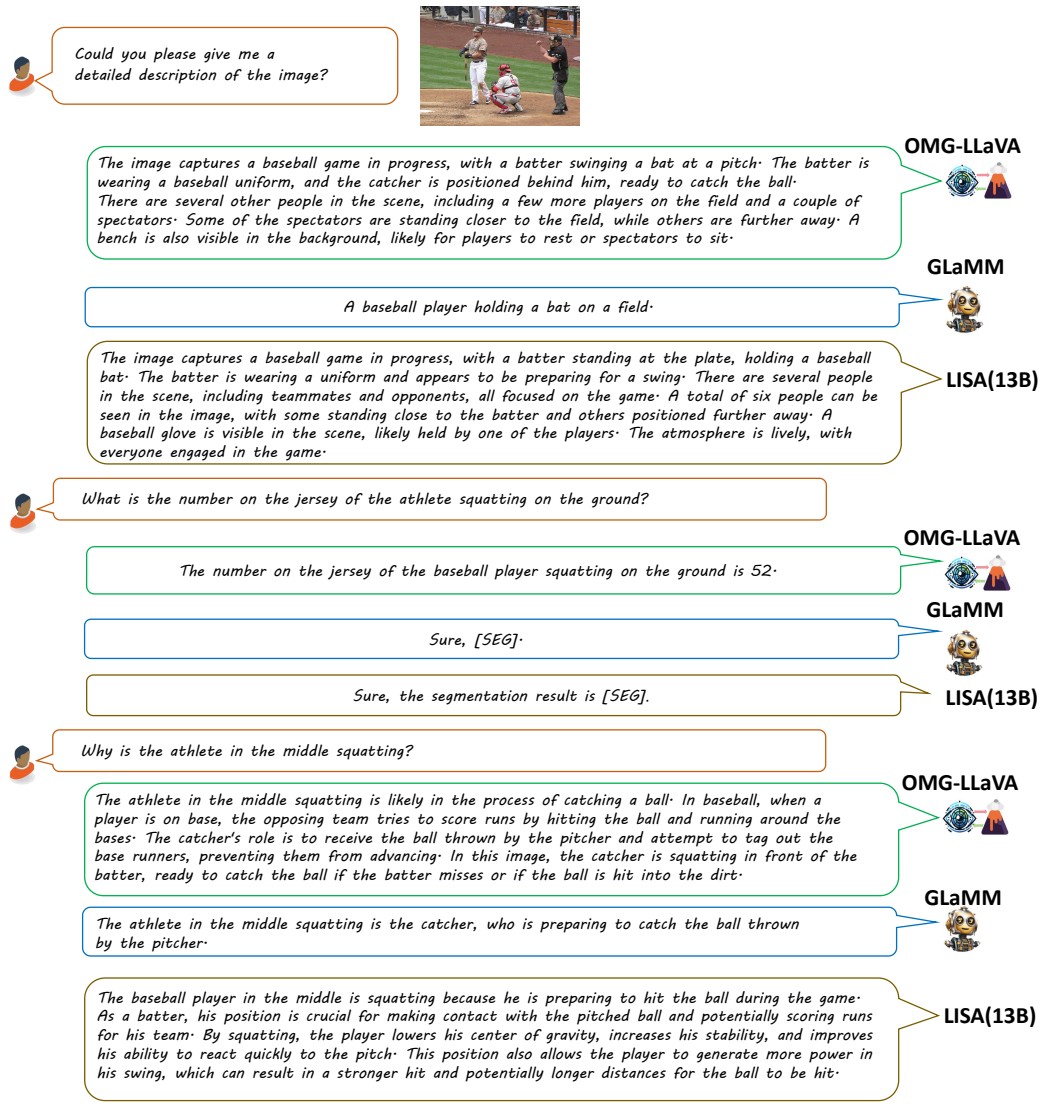

Figure 10: Qualitative comparison on the image-based conversation task.

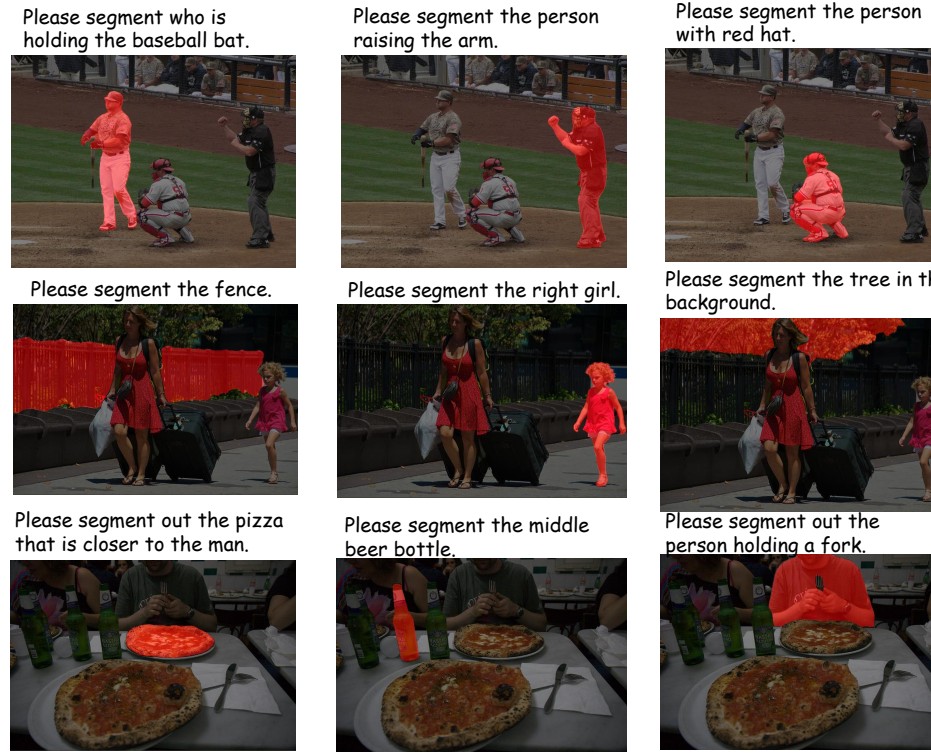

Figure 11: More visualization results of referring expression segmentation.

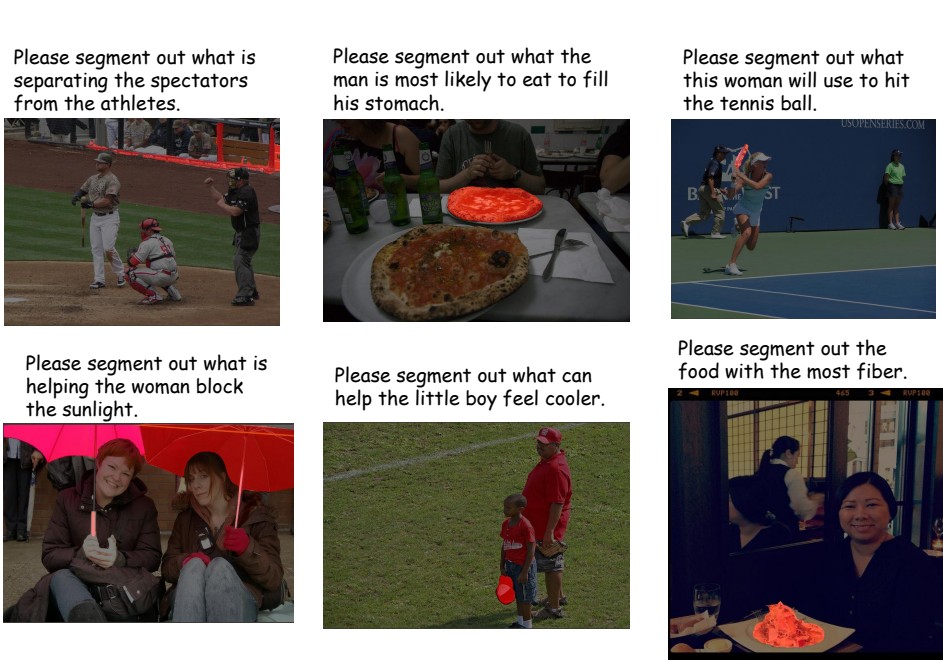

Figure 12: Visualization results of reasoning segmentation.

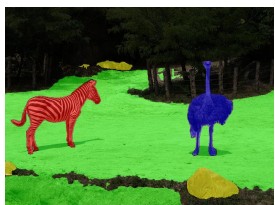 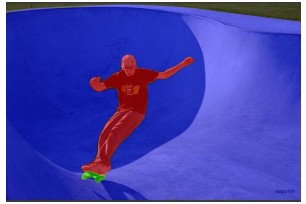 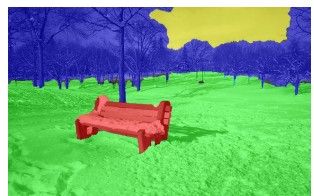

*The image shows a zebra standing on dirt. A bird is also present on the dirt. The dirt is attached to a rock.*

*A skateboarder is grinding his board on a ramp*

*The image shows a bench situated on snow. The bench is located in front of a tree. The sky is visible over the tree.*

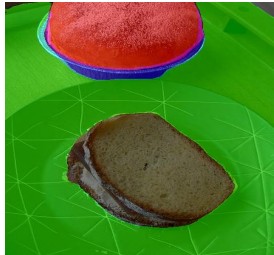 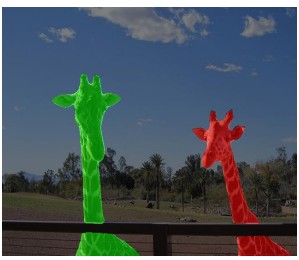 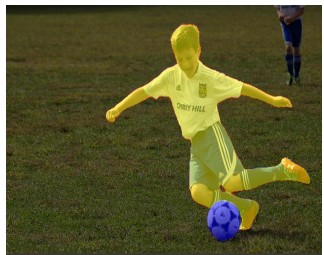

*The image shows a cake placed on a dining table. The cake is attached to a bowl.*

*a giraffe standing behind another giraffe. a giraffe standing in front of another giraffe*

*A boy in a blue uniform kicks a soccer ball*

Figure 13: More visualization results of grounded conversation generation.

Point-prompt      Box-prompt      Mask-prompt

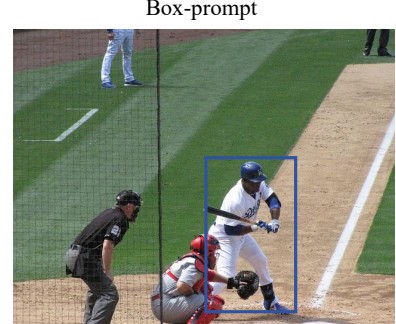 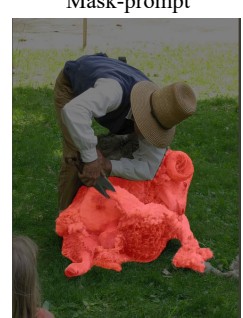

The purple umbrella in the image is a large umbrella, providing protection from the falling snow. It is held by the man, who is walking alongside the woman. The umbrella is likely waterproof, allowing them to stay dry while they walk through the snow.

The athlete wearing a blue helmet is the batter in the baseball game. He is in the process of swinging the bat, attempting to hit the ball that is coming towards him. The batter is likely focused on the ball's trajectory and timing his swing to make contact with it.

The sheep in the image is a large, wooly animal with a thick coat of wool. It is lying down on the grass, appearing to be relaxed and comfortable. The sheep is being sheared by a man, who is carefully removing the wool from its body.

Figure 14: More visualization results of grounded conversation generation.

