# OpenReview forum: "OMG-LLaVA: Bridging Image-level, Object-level, Pixel-level Reasoning and Understanding"
_NeurIPS.cc/2024/Conference — NeurIPS 2024 poster_

### Official Review · Reviewer_UauW · 2024-07-05

**Soundness:** 4
**Presentation:** 3
**Contribution:** 4
**Rating:** 8
**Confidence:** 5

**Summary:**

This work presents a new multi-modal large language model, named OMG-LLaVA. This model builds on previous work OMG-Seg and combines a LLM (LLaVA-like) into one simple framework. Compared with previous MLLMs, this work unifies lots of image-level, object-level, and pixel-level segmentation and reasoning tasks in one shot.

**Strengths:**

1, The motivation and goal are interesting and ambitious. The proposed architecture is novel and interesting to me. Although built from previous universal segmentation network, the authors show a simple but effective way to connect LLM and dense visual perception model.
2, The proposed method only uses one image encoder and one image decoder, which is clean and easy to follow. Moreover, the authors propose a new connection module, named perception prior module, which also works effective in connecting object query into LLM for further processing.
3, Compared to several works, including PixelLM and GLaMM, OMG-LLaVA has comprehensive functionality but better performance. Moreover, the entire design is simple and elegant.
4, Overall writing is good and is easy to follow.

**Weaknesses:**

Overall, I think this paper is good and elegant, compared with previous combined approach. However, there are several details should be added for the better draft.
1, More detailed designs or ablation should be carried out for perception prior embedding design. The current draft only shows the effectiveness of proposed approach.

2, More detailed discussion on meta-architecture design should be
One question is why you fix the image decoder why not use another decoder with the same architecture but with the same pre-trained weights.
Although this operation can increase the parameter costs, I wonder whether joint sharing one decoder can have mutual effects.
Thus, more detailed experiments on meta-architecture design should be added.
3, No parameter and Gflops analysis. For example, the author claim they only use one image encoder, one decoder and one LLM. Compared with previous combined approach (GLaMM), this advantage is not well explored.

**Questions:**

See the weakness, I would raise the score if all questions are well solved.

**Limitations:**

Please see weaknesses.

---

> ### Author Rebuttal · Authors · 2024-08-06
>
> **Q1. More detailed designs or ablation should be carried out for perception prior embedding design.**
>
> **A**: Thanks for your suggestion. We have added more detailed ablation studies on perception prior embedding strategy. We have conducted ablation studies on different strategies for assigning perception embeddings to each pixel token, and the results are shown in Table R5. The best results were achieved by assigning the corresponding object query to each pixel as the perception embedding to be embedded, using the SoftMax method. The strategy of generating perception embeddings through L1 normalization led to performance degradation, but it still brought a significant performance improvement compared to not using the perception prior embedding strategy.
>
> ---
>
> **Q2. More detailed discussion on meta-architecture design should be One question is why you fix the image decoder why not use another decoder with the same architecture but with the same pre-trained weights. Although this operation can increase the parameter costs, I wonder whether joint sharing one decoder can have mutual effects. Thus, more detailed experiments on meta-architecture design should be added.**
>
> **A**: Using another decoder with the same architecture and loaded with the same pre-trained weights to generate segmentation results in a marginal performance improvement, as shown in the table below. Undoubtedly, using an additional decoder can achieve better performance due to the introduction of more trainable parameters. However, the performance gain is marginal, so we chose to adopt a frozen OMG Decoder to achieve a balance between performance and parameters.
>
> | Decoder  |      | refCOCO |       |      | refCOCO+ |       | refCOCOg |      |        | GCG   |      |      |
> | -------- | ---- | ------- | ----- | ---- | -------- | ----- | -------- | ---- | ------ | ----- | ---- | ---- |
> |          | Val  | TestA   | TestB | Val  | TestA    | TestB | Val      | Test | METEOR | CIDEr | AP50 | mIOU |
> | unfrozen | 78.0 | 80.3    | 74.1  | 69.1 | 73.1     | 63.0  | 72.9     | 72.9 | 14.5   | 38.5  | 28.6 | 64.7 |
> | frozen   | 77.2 | 79.8    | 74.1  | 68.7 | 73.0     | 61.6  | 71.7     | 71.9 | 14.5   | 37.5  | 28.9 | 64.6 |
>
> ---
>
> **Q3. No parameter and Gflops analysis. For example, the author claim they only use one image encoder, one decoder and one LLM. Compared with previous combined approach (GLaMM), this advantage is not well explored.**
>
> **A**: We have statistics on the parameters of the main components of OMG-LLaVA, such as the LLM and visual encoder, and the results are shown in below table. The visual encoder of OMG-LLaVA is only 0.2B, much smaller than the 0.9B of LISA and GLaMM. The number of vision tokens significantly impacts the computational cost of the LLM. Compared to GLaMM, which uses 576 visual tokens, OMG-LLaVA only requires about 276 visual tokens, which is only 50% of GLaMM.
>
> | Methods   | LLM  | Visual Encoder             | Image Size                 | Visual Tokens                            |
> | --------- | ---- | -------------------------- | -------------------------- | ---------------------------------------- |
> | LISA      | 7B   | VIT-L(0.3B) \& VIT-H(0.6B) | (224, 224) \& (1024, 1024) | 256                                      |
> | GLaMM     | 7B   | VIT-L(0.3B) \& VIT-H(0.6B) | (336, 336) \& (1024, 1024) | 576                                      |
> | OMG-LLaVA | 7B   | ConvNext-L(0.2B)           | (1024, 1024)               | 256(Pixel-centric) \& 20(Object-centric) |

---

> ### Comment · Reviewer_UauW · 2024-08-09
>
> After going through the feedback from other reviewers and the author's detailed replies, I'm more convinced this paper lives up to the NeurIPS standards. I’m especially impressed by the authors’ thorough and detailed response, with the solid experimental results they added. They addressed the key issues I brought up. So, I've bumped my score up to 8, with a clear accept.
>
> Just to sum up my thoughts on this research: The paper introduces OMG-LLaVA, which is a trailblazer in merging three levels of perception into a single LLM system, using just one encoder, one decoder, and one LLM. The way the authors link object queries from a universal segmentation model to the LLM is really smart. It's not just about beating other models like Pixel-LLM and GLaMM—it’s about doing it more elegantly. During the rebuttal, the authors also showed that OMG-LLaVA performs even better with a stronger LLM, pointing to great potential for future work.
>
> At last, the authors should open-source the entire codebase and models (including the stronger LLMs in the rebuttal) for the community soon.

---

> > ### Author Response · Authors · 2024-08-11
> > **Thanks for the response.**
> >
> > Thanks for your comments and acknowledgement to our work.
> >
> > We will release the entire codebase, including training, testing, and demo model.
> >
> > Best regards!
> >
> > Authors of 8973

---

### Official Review · Reviewer_Timw · 2024-07-11

**Soundness:** 3
**Presentation:** 3
**Contribution:** 3
**Rating:** 7
**Confidence:** 4

**Summary:**

This paper proposes OMG-LLaVA, an elegant MLLM framework. OMG-LLaVA achieves pixel-level, object-level, and image-level understanding and reasoning with only a perception model and a language model. OMG-LLaVA exhibits more comprehensive capabilities than current MLLMs, such as LLaVA, Osprey, and GLaMM. The experimental results on several benchmarks demonstrate that OMG-LLaVA achieves performance comparable to SOTA methods.

**Strengths:**

1 The motivation is very clear and easy to follow.
2 The OMG-LLaVA is a simpler and more efficient MLLM framework compared to current MLLMs, and it has broader capabilities such as universal image segmentation and supporting comprehensive visual prompt inputs.
3 The proposed perception prior embedding strategy is crucial and efficient for providing perception results for LLMs. This approach allows LLMs to achieve grounded outputs similar to selecting object tokens from inputs, which is more reasonable and performs better on segmentation tasks than LISA, PixelLM, and GLaMM.
4 OMG-LLaVA unifies the visual prompt embeddings and segmentation embeddings as object-centric visual tokens, effectively shortening the training phases.

**Weaknesses:**

1 The paper needs to include more ablation experiments to help readers better understand the work. For example, what would happen if the visual projector design shared an MLP for pixel- and object-centric visual tokens?
2 Would the performance of OMG-LLaVA improve with a larger LLM?
3 The authors should include more MLLMs in Table 1 for a more comprehensive comparison.

**Questions:**

1 The paper needs to include more ablation experiments to help readers better understand the work. For example, what would happen if the visual projector design shared an MLP for pixel- and object-centric visual tokens?
2 Would the performance of OMG-LLaVA improve with a larger LLM?

**Limitations:**

The authors pointed out the limitations of OMG-LLaVA, including performance degradation on image-level tasks and lack of capability for part-level segmentation. The latter can be addressed by replacing the perception model with a more powerful one.

---

> ### Author Rebuttal · Authors · 2024-08-06
>
> **Q1. The paper needs to include more ablation experiments to help readers better understand the work. For example, what would happen if the visual projector design shared an MLP for pixel- and object-centric visual tokens?**
>
> **A:** Table R4 presents the results of ablation studies on the projector. We observed a substantial performance degradation when pixel-centric and object-centric tokens shared a projector. This phenomenon can be attributed to the distinct semantic and perception information encoded in these two types of tokens. Sharing a projector results in detrimental interference between them. Furthermore, our experiments with additional cross-attention layers in the projector did not yield any performance benefits. For the object-centric tokens, specifically visual prompt tokens and object queries, sharing a projector resulted in marginal performance gains for both segmentation and visual prompt understanding tasks.
>
> ---
>
> **Q2. Would the performance of OMG-LLaVA improve with a larger LLM?**
>
> **A:** We have conducted experiments with OMG-LLaVA using various LLMs, and the results are presented in Table R3. Our findings indicate that the performance of OMG-LLaVA increases with the strength of the LLM. Due to time limitations, the performance of OMG-LLaVA models based on larger LLMs, such as Yi-34b, will be reported in the coming days.
>
> ---
> **Q3. The authors should include more MLLMs in Table 1 for a more comprehensive comparison.**
>
> **A:** We appreciate your feedback. To further emphasize the unique capabilities of OMG-LLaVA, we have compared more MLLMs (about 25 MLLMs). The result demonstrates that OMG-LLaVA provides the most comprehensive capabilities while employing a single visual encoder.

---

> > ### Comment · Reviewer_Timw · 2024-08-12
> > **Concerns Addressed**
> >
> > Thanks for the further explanation. Please include the supplementary experiments in the final version of the submission. I'll keep my positive rating.

---

### Official Review · Reviewer_ck7P · 2024-07-11

**Soundness:** 3
**Presentation:** 3
**Contribution:** 2
**Rating:** 4
**Confidence:** 5

**Summary:**

This paper presents OMG-LLaVA, which facilitates pixel-level, object-level and image-level understanding tasks within a unified framework. Within the OMG-LLaVA, an OMG decoder and perception prior embedding approach are proposed to enhance object-centric comprehension. Comparison with state-of-the-art methods prove the effectiveness of the proposed method.

**Strengths:**

1. This paper integrates several segmentation tasks and object-level visual comprehension ability within a unified framework. Using a high-resolution image encoder and compressing the visual tokens to save computation resources is reasonable.
2. The design of perception prior embedding is interesting which can integrate object-level cues with visual tokens.

**Weaknesses:**

1. The proposed method is not very elegant as too many designs customized for different tasks are introduced in the framework. For different tasks (e.g., image captioning, image segmentation and visual prompt based comprehension), different workflows should be conducted.
2. Some important technical details are omitted. For example, to encourage different object queries to attend to different objects, hungarian matching algorithm should be applied to assign labels for each individual object query, especially for COCO segmentation where multiple objects are supposed to be simultaneously segmented. Besides, to generate masks from object queries, the mask decoder should be carefully designed to generate high-quality masks, however, this paper only mentioned a simple FFN is utilized. Is it reasonable? More details should be provided.
3. Some important experimental results are not provided. For example, experimental results on prevalent VQA benchmarks like MMBench, MME, SEEDBench, etc. should be provided to illustrate the general-purpose conversational capability of OMG-LLava. Whether introducing these segmentation data can hurt the VQA capabilities of the model? Overall, only segmentation and simple captioning qualitative results are reported, which are much less than the supported functions of the proposed OMG-LLava.

**Questions:**

Since comprehension capabilities ranging from image-level to object-level are established, does OMG-LLava possess composite capabilities such as visualizing the intention of LLM during a conventional conversation in the form of segmentation masks? This is very helpful of understanding the working mechanism of LLMs.

**Limitations:**

See weakness.

---

> ### Author Rebuttal · Authors · 2024-08-06
>
> **Q1. The proposed method is not very elegant as too many designs customized for different tasks are introduced in the framework.**
>
> **A**: OMG-LLaVA boasts an elegant model architecture and more streamlined workflows across tasks than GLaMM. For instance, in GLaMM, visual prompt encoding and segmentation mask decoding employ entirely different workflows and models. However, in OMG LLaVA, visual prompt encoding and segmentation mask decoding are unified into a single OMG Decoder to generate object-centric queries. Maintaining a consistent workflow while unifying many tasks is extremely challenging and currently difficult. For example, mask decoding and visual prompt encoding are hard to implement solely with LLMs without relying on other modules. We believe that in the future, diverse tasks will be unified under elegant architectures and shared workflows, and OMG-LLaVA is a significant step toward this goal.
>
> ---
>
> **Q2. Some important technical details are omitted. For example, to encourage different object queries to attend to different objects, a Hungarian matching algorithm should be applied to assign labels for each individual object query, especially for COCO segmentation, where multiple objects are supposed to be simultaneously segmented. Besides, to generate masks from object queries, the mask decoder should be carefully designed to generate high-quality masks, however, this paper only mentioned a simple FFN is utilized. Is it reasonable? More details should be provided.**
>
> **A**: There is a misunderstanding. The segmentation ability is derived from the frozen perception model (OMG-Seg), not the LLM's output. OMG-Seg is a component of OMG-LLaVA, so there is no need for the LLM to reiterate these perception results.
> Secondly, the mask decoder of OMG-LLaVA is not a simple FFN layer. The last layer's hidden states of the [SEG] token first pass through a text projector (an MLP layer) and then are fed into the frozen mask decoder (OMG-Seg head) to obtain segmentation masks. During training, the mask decoder remains frozen while the text projector is trained. Since the perception results have already been embedded into the LLM's input via perception prior embedding, generating segmentation queries is not difficult. As shown in Table R4 in our PDF file, we also experimented with adding extra cross-attention layers in the text projector, but the performance was similar to using a simple MLP.
>
> ---
>
> **Q3. Some important experimental results are not provided. For example, experimental results on prevalent VQA benchmarks like MMBench, MME, SEEDBench, etc. should be provided to illustrate the general-purpose conversational capability of OMG-LLava.**
>
> **A**: We have evaluated the performance of OMG-LLaVA on multiple image-level benchmarks, including MMbench, MME, SEED-Bench, POPE, and AI2D. The results are presented in Table R2. When jointly co-trained with both image-level and segmentation-based data, OMG-LLaVA significantly outperformed LISA and GLaMM on all image-level benchmarks. However, co-training with segmentation-based data significantly decreases performance on image-level benchmarks, although OMG-LLaVA can mitigate this issue to a large extent through perception prior embedding. We also evaluate the performance of OMG-LLaVA trained solely on the LLaVA dataset on image-level benchmarks. OMG-LLaVA outperformed LLaVA 1.5 on MME, SEED-Bench, POPE, and AI2D by 41, 3.0, 3.0, and 5.1 points, respectively.
>
> ---
>
> **Q4. Since comprehension capabilities ranging from image-level to object-level are established, does OMG-LLava possess composite capabilities such as visualizing the intention of LLM during a conventional conversation in the form of segmentation masks? This is very helpful of understanding the working mechanism of LLMs.**
>
> **A**: As shown in Figure R6, OMG-LLaVA can visualize LLM intentions through segmentation masks. While the ability to visualize LLM intentions in diverse conversations was not explicitly included in OMG-LLaVA's training data, the model learned this capability by training on referring expression segmentation data and image-level conversation data. This ability is crucial for AI assistants, as it significantly enhances the user experience.

---

> > ### Comment · Reviewer_ck7P · 2024-08-09
> > **Reply to authors rebuttal**
> >
> > **R1: Framework Design**
> >
> > I disagree with the author's claim that *"mask decoding and visual prompt encoding are hard to implement solely with LLMs without relying on other modules"*, because there are already related works that handle mask decoding and visual prompts using the LLM alone. Specifically, VisionLLM [1] transfers polygon masks into a sequence of points sampled from their contour as LLM's learning target. ViP-LLaVA [2] renders the visual prompts in the image, and feeds the image into the LLM without any visual prompt encoding techniques for training.
> >
> > [1] Wang W, Chen Z, Chen X, et al. Visionllm: Large language model is also an open-ended decoder for vision-centric tasks[J]. Advances in Neural Information Processing Systems, 2024, 36.
> >
> > [2] Cai M, Liu H, Mustikovela S K, et al. ViP-LLaVA: Making Large Multimodal Models Understand Arbitrary Visual Prompts[C]//Proceedings of the IEEE/CVF Conference on Computer Vision and Pattern Recognition. 2024: 12914-12923.
> >
> > **R2: Technical Details**
> >
> > My major problem is how the set of learnable object queries in the OMG decoder can automatically capture distinct object regions? As a comparison, each object query in the DETR should be assigned a unique GT box as the supervision. This is crucial in establishing the dense instances perception capability.
> >
> > Since OMG-Seg, a powerful foundation model that can solve universal segmentation tasks, is adopted as the mask decoder, the expression that *"the object queries can be decoded into segmentation masks and object categories via a simple FFN layer"* is somewhat misleading.
> >
> > **R3: VQA Results**
> >
> > Thanks for authors' efforts, I have another follow-up question about the provided VQA results.
> >
> > 1. What makes the OMG-LLaVA perform better than LLaVA-v1.5 using the same training data?
> >
> > **Follow-up questions:**
> >
> > Now that the OMG-LLaVA has object-level perception ability, can OMG-LLaVA tackle traditional task like object detection?

---

> > > ### Author Response · Authors · 2024-08-10
> > >
> > > Thanks for your efforts and quick response.
> > >
> > > > **R1:  Framework Design**
> > >
> > > We apologize for the previous imprecise statement. We would like to rephrase it as "**Fine-grained** mask decoding and **flexible** visual prompt encoding are challenging to implement solely with LLMs without relying on additional modules."
> > >
> > > While we acknowledge that VisionLLM and VIP-LLaVA can respectively achieve mask decoding and visual prompt encoding without external modules, **these approaches have significant limitations**. For instance, VisionLLM decodes N contour points similar to PolarMask[1] to generate segmentation results, **but contour-based representations inherently suffer from inaccuracies** when compared to binary masks for precise object segmentation. This makes it hard to segment objects with hole. OMG-LLaVA adopt query-based baseline, OMG-Seg as baseline, which can avoid this issue. VIP-LLaVA encodes visual prompts by drawing them onto the image, **requiring all visual prompts to be provided before the dialogue**, limiting the flexibility to add new prompts during the conversation. OMG-LLaVA, on the other hand, leverages the OMG decoder to produce high-quality binary mask segmentation results and flexibly decode user-input visual prompts during the conversation.
> > >
> > > [1] Xie E, Sun P, Song X, et al. Polarmask: Single shot instance segmentation with polar representation. CVPR 2020.
> > >
> > > ***
> > >
> > > > **R2: Technical Details**
> > >
> > > OMG-LLaVA employs a pre-trained OMG-Seg model as its visual encoder. Similar to DETR, OMG-Seg utilizes learnable queries and has been pre-trained on the COCO dataset. The training of OMG-Seg undoubtedly requires the Hungarian matching algorithm to assign labels akin to DETR. However, in the training of OMG-LLaVA, OMG-Seg is kept frozen, eliminating the need for training and Hungarian matching. Instead, it performs the inference process of panoptic segmentation. The dense instance perception capability is derived from the pre-trained OMG-Seg.
> > >
> > > The statement "the object queries can be decoded into segmentation masks and object categories via a simple FFN layer" is a detail describe of the OMG-Seg model and may lead to some misunderstandings. In OMG-LLaVA, the hidden states of the [SEG] token are transformed into initial object queries through an MLP projector. Subsequently, these initial queries are fed into the OMG decoder, where they interact with image features via self- and cross-attention layers to produce final object queries. Within the OMG decoder, the final object queries are passed through a simple FFN layer to obtain segmentation masks and object categories. We will rephase this part in the next draft.
> > >
> > > ***
> > >
> > > > **R3: VQA Results**
> > >
> > > When trained exclusively on the LLaVA dataset, the primary distinction between OMG-LLaVA and LLaVA 1.5 lies in the perception prior embedding. Despite employing the same LLM, an identical MLP projector, and similar CLIP backbones (ConvNext-L for OMG-LLaVA and ViT-L for LLaVA 1.5), the incorporation of perception results as input to the LLM enables a more profound understanding of images. Analogous findings have been reported in Vcoder [2] and SeTok [3].
> > >
> > > [2] Jain J, Yang J, Shi H. Vcoder: Versatile vision encoders for multimodal large language models. CVPR 2024.
> > >
> > > [3] Wu S, Fei H, Li X, et al. Towards Semantic Equivalence of Tokenization in Multimodal LLM. ArXiv, 2024.
> > >
> > > ***
> > >
> > > > R4: Now that the OMG-LLaVA has object-level perception ability, can OMG-LLaVA tackle traditional task like object detection?
> > >
> > > OMG-Seg is capable of handling both instance segmentation and panoptic segmentation, and bounding boxes can be easily extracted from the generated binary masks. As OMG-LLaVA incorporates a pre-trained OMG-Seg, it inherently possesses all of OMG-Seg's capabilities, such as object detection. OMG-LLaVA achieves 44.5 mAP on the COCO instance segmentation benchmark and 45.8 mAP on the COCO object detection benchmark.
> > >
> > > In OMG-LLaVA, current dense segmentation tasks such as panoptic segmentation rely solely on OMG-Seg and do not utilize the LLM. We think that training the LLM to reproduce the perception results already output by OMG-Seg is unnecessary and meaningless, even though it could be achieved through simple training.

---

> > > > ### Author Response · Authors · 2024-08-11
> > > > **Please let us know whether we address all the issues**
> > > >
> > > > Dear reviewer,
> > > >
> > > > Thank you for the comments on our paper.
> > > >
> > > > We have submitted the response to your comments. Please let us know if you have additional questions so that we can address them during the discussion period. We hope that you can consider raising the score after we address all the issues.
> > > >
> > > > Thank you

---

> > > ### Author Response · Authors · 2024-08-12
> > > **Please let us know whether we address all the issues**
> > >
> > > Dear reviewer,
> > >
> > > Has our response addressed your questions? If you have any other questions or feel that some issues have not been adequately answered, please let us know. If not, we kindly ask you to consider raising your score.
> > >
> > > Thank you

---

> > > > ### Author Response · Authors · 2024-08-14
> > > > **Please let us know whether we address all the issues**
> > > >
> > > > Dear reviewer #ck7P,
> > > >
> > > > We kindly ask again whether all your questions have been resolved, as the rebuttal deadline is approaching. If they have been resolved, please raise your score.
> > > >
> > > > Thank you

---

> > > > > ### Comment · Reviewer_ck7P · 2024-08-14
> > > > > **Reply to authors rebuttal**
> > > > >
> > > > > Thanks for authors further rebuttal.
> > > > >
> > > > > While I acknowledge that learning masks as a sequence of coordinates is generally less effective than directly predicting them, it appears that the pixel-level and object-level understanding capabilities primarily stem from the robust vision foundation model, OMG-Seg, rather than from the MLLM itself. This integration adds an extra vision encoder and decoder from the foundation model, which significantly increases the computational load compared to a pure MLLM. Given the use of such a powerful visual foundation model, why not simply address downstream tasks like segmentation by utilizing a specialist model's API, as is done in HuggingGPT, if pursuing a higher performance?
> > > > >
> > > > > Overall, I believe the contribution of the proposed model is diminished by the reliance on the powerful OMG-Seg model. I've raised my score to 4, pending further discussion with the PC and ACs.

---

### Official Review · Reviewer_czBJ · 2024-07-13

**Soundness:** 2
**Presentation:** 2
**Contribution:** 2
**Rating:** 5
**Confidence:** 4

**Summary:**

This work proposes, OMG-LLaVA, a unified framework for image-level, object-level, and pixel-level vision-language comprehension. In particular, OMG-Seg, a universal image segmentation model, is integrated with a LLaVA-like multimodal large language model (MLLM), so that various image-level (e.g., image captioning, visual question answering), object-level (e.g., promptable segmentation, region captioning), and pixel-level (e.g., referring expression segmentation, grounded conversation generation) vision-language tasks can be performed. By using one unified visual encoding and decoding module for objects and pixel masks, OMG-LLaVA poses a simple design and shows competitive performance compared with prior models.

**Strengths:**

1. Overall, OMG-LLaVA’s architecture design is more unified and concise when compared to prior models like LISA and GLaMM where more than one visual encoder is applied.

2. OMG-LLaVA is able to perform a wide range of vision-language comprehension tasks with one model.

3. The method description is generally clear and easy to follow.

**Weaknesses:**

1. [Performance] Although this work claims OMG-LLaVA as a generalist model, it cannot outperform prior state-of-the-art models on any specific task. This performance difference would weaken the applicability. In Table 3, the performance should also be compared with GLaMM (which has significantly better performance than LISA).

2. [Technical contribution] Overall, OMG-LLaVA seems a direct integration of OMG-Seg and LLaVA. The most novel design in this work seems to be the perception prior embedding, which enables the connection between OMG-Seg and the object visual tokens in LLaVA. The technical contribution is thus somewhat limited. It would be great if the authors could better discuss the novel technical contributions of this work.

3. [Image-level tasks] Although this work claims OMG-LLaVA can unify image-level, object-level, and pixel-level reasoning and understanding tasks, there is no quantitative evaluation in terms of image-level tasks such as VQA.

4. [Frozen perception module] In OMG-LLaVA, the “universal perception module” (OMG-Seg) is frozen (Figure 2e) to preserve its learned knowledge. However, Table 3 shows that allowing the OMG decoder to be tuned can actually improve the referring segmentation performance. It is not explained why this module should be frozen in other tasks, or whether this practice would bring further gains.

**Questions:**

In addition to the weaknesses mentioned above, there are two minor questions:

1. What is the “pixel shuffle operator” (Line 171)?

2. To be consistent with Equation 2, should the segmentation mask be written as $\mathcal{M}\in\mathbb{R}^{HW\times N_q}$ instead of $\mathcal{M}\in\mathbb{R}^{N_q\times HW}$?

**Limitations:**

The authors have discussed the technical limitations, but have not discussed potential societal impacts. There is no explanation about why this work has no societal impacts, either.

---

> ### Author Rebuttal · Authors · 2024-08-06
>
> **Q1. [Performance] Although this work claims OMG-LLaVA as a generalist model, it cannot outperform prior state-of-the-art models on any specific task.**
>
> **A**: We have updated the performance of OMG-LLaVA, as shown in Table R1. OMG-LLaVA outperforms LISA, PixelLM, and GSVA on the RES benchmarks. GLaMM used the GranD dataset for pretraining. The GranD pretrain dataset contains 11M images, far exceeding the GranD-f dataset used by OMG-LLaVA, which only includes 210K (0.21M) images. Nevertheless, OMG-LLaVA still surpasses GLaMM on the GCG benchmark by 0.6 CIDEr, 1.4 AP50, and 0.1 mIoU.
>
> Additionally, OMG-LLaVA achieves substantial performance gains over GLaMM, LISA, and PixelLM on image-level benchmarks (Table R2). For instance, when jointly co-training with image-level and segmentation-based datasets, OMG-LLaVA achieves 1412 on MME benchmark, outperforming LISA, PixelLM and GLaMM with 1410, 968 and 1389, respectively. When only trained on the LLaVA dataset, OMG-LLaVA outperforms LLaVA-1.5 by impressive margins of 41, 3.0, 3.0, and 5.1 points on MME, SEED-Bench, POPE, and AI2D, respectively.
>
> ---
>
> **Q2. [Technical contribution] Overall, OMG-LLaVA seems a direct integration of OMG-Seg and LLaVA.**
>
> **A**: Our most significant contribution is introducing an elegant MLLM architecture that comprises only a perception model and an LLM, capable of understanding and reasoning at the image, object, and pixel levels. The frozen perception decoder can simultaneously encode visual prompts and decode [SEG] tokens. However, our experiments and those in LISA have shown that the frozen perception decoder cannot effectively decode [SEG] tokens. The proposed perception prior embedding is the key component enabling our meta-architecture to work well. Thus, we have three main contributions.
>
> - First, we propose an elegant MLLM architecture with a single visual encoder, a perception head, and an LLM, which can simultaneously handle image, object, and pixel-level tasks.
>
> - Secondly, we introduce perception prior embedding to help the LLM better align pixel-centric visual tokens with object-centric visual tokens, allowing the LLM to generate object-centric tokens that the frozen perception decoder can decode based on the input embedded perception prior.
>
> - Thirdly, we construct OMG-LLaVA, which achieves SOTA performance on pixel-level tasks and comparable accuracy to SOTA on object-level tasks. Additionally, OMG-LLaVA significantly outperforms versatile MLLMs like LISA, PixelLM, and GLaMM in image-level tasks.
>
> From these aspects, we argue that our work is not the direct integration of OMG-Seg and LLaVA.
>
> ---
>
> **Q3. [Image-level tasks] Although this work claims OMG-LLaVA can unify image-level, object-level, and pixel-level reasoning and understanding tasks, there is no quantitative evaluation in terms of image-level tasks such as VQA.**
>
> **A**: Table R2 presents our model's performance on various image-level benchmarks, including MME, MMBench, SEED-Bench, POPE, and AI2D. Jointly co-training on segmentation and image-level data can detrimentally impact an MLLM's performance on image-level benchmarks. However, OMG-LLaVA mitigates this negative impact thanks to its perception prior embedding. OMG-LLaVA significantly outperforms GLaMM, PixelLM, LISA, and LaSagnA on these image-level benchmarks. OMG-LLaVA scores 1412 on MME and 47.7 on MMBench, surpassing GLaMM with 1389 on MME and 11.1 on MMBench. By training solely on image-level data, OMG-LLaVA achieves superior performance over LLaVA 1.5 on benchmarks such as MME, SEED-Bench, POPE, and AI2D, attributed to its perception prior embedding.
>
> ---
>
> **Q4. [Frozen perception module] In OMG-LLaVA, the “universal perception module” (OMG-Seg) is frozen (Figure 2e) to preserve its learned knowledge. However, Table 3 shows that allowing the OMG decoder to be tuned can actually improve the referring segmentation performance. It is not explained why this module should be frozen in other tasks, or whether this practice would bring further gains.**
>
> **A**: During finetuning, the OMG decoder is duplicated and unfrozen to better decode the [SEG] tokens generated by the LLM. The encoding of object-centric tokens and visual prompts continues to be handled by the original frozen OMG decoder. We also add the experiment of keeping the OMG decoder frozen when finetuning the model. As shown in the last two lines of Table R1, finetuning on the RES datasets while keeping the OMG decoder frozen results in a slight performance degradation compared to the case where an additional OMG decoder is duplicated and unfrozen (78.0 mIoU *vs.* 77.2 mIoU). However, the performance still significantly surpasses LISA and PixelLM. Moreover, we directly unfreeze the original OMG decoder without duplicating it. In that case, OMG-LLaVA cannot work well due to the reliance of object-centric tokens on a stable mask segmentation capability.
>
> ---
>
> **Q5. What is the “pixel shuffle operator” (Line 171)?**
>
> **A**: This paper leverages the pixel shuffle operator to downsample image features. By flattening $S \times S$ neighboring pixel features, the operator reduces the image size from $(H, W, C)$ to $(H/S, W/S, C \times S^{2})$.
>
> ---
>
> **Q6. To be consistent with Equation 2.**
>
> **A**: Thank you for the suggestion. We have modified $M \in R^{N_{q}\times HW}$ to $M \in R^{HW \times N_{q}}$.

---

> > ### Author Response · Authors · 2024-08-12
> > **Please let us know whether we address all the issues**
> >
> > Dear reviewer,
> >
> > Thank you for the comments on our paper.
> >
> > We have submitted the response to your comments and a PDF file. Please let us know if you have additional questions so that we can address them during the discussion period. We hope that you can consider raising the score after we address all the issues.
> >
> > Thank you

---

> > > ### Comment · Area_Chair_hgYa · 2024-08-13
> > >
> > > Dear reviewer,
> > >
> > > Please carefully read the rebuttal and the other reviews, then reply to the authors indicating whether your questions/concerns have been addressed. If not, please specify which questions/concerns remain unresolved so that the authors have a fair opportunity to address them before the author-reviewer discussion period ends in approximately 36 hours from now (August 13th, 11:59pm AoE).
> > >
> > > Best,
> > > AC

---

> > > > ### Comment · Reviewer_czBJ · 2024-08-13
> > > >
> > > > Thanks to the authors for their detailed response. In their response, the performance of OMG-LLaVA is updated, including a more comprehensive evaluation on image-level benchmarks with or without segmentation data tuning. The technical contribution is sufficiently clarified. Therefore, I would like to raise the rating to 5.

---

> > > > > ### Author Response · Authors · 2024-08-14
> > > > > **Response to Reviewer czBJ**
> > > > >
> > > > > Thanks for your reply. We will merge your comments and extended experiment results into our next draft.
> > > > >
> > > > >
> > > > > Best regards!
> > > > >
> > > > > Authors of OMG-LLaVA

---

### Author Rebuttal · Authors · 2024-08-06

# General Responses

---
Dear Reviewers,

We thank all the reviewers for the detailed suggestions. All reviewers acknowledge the technical contributions of our work, including the new unified designs for MLLM and comprehensive benchmark evaluation. We listed additional important experiments and answered common questions here. We will respond individually to the corresponding reviewers for the detailed questions of each reviewer.

---

## 1. Evaluation on image-level benchmarks (R#czBJ and R#ck7P)

We have evaluated the performance of OMG-LLaVA on multiple image-level benchmarks, including MMbench, MME, SEED-Bench, POPE, and AI2D. The results are presented in Table R2 (please refer to the submitted pdf). When jointly co-trained with both image-level and segmentation-based data, OMG-LLaVA significantly outperformed LISA, GLaMM, PixelLM, and LaSagnA on all image-level benchmarks. For example, OMG-LLaVA scores 1257 on MME and 45.7 on MMBench, surpassing GLaMM with 1234 on MME and 8.9 on MMBench. We list these results in the pdf file.

However, co-training with segmentation-based data significantly decreases performance on image-level benchmarks (This is also verified in previous work, PixelLLM (https://arxiv.org/pdf/2312.02228), see the last page of their arxiv.), although OMG-LLaVA can mitigate this issue to a large extent through perception prior embedding. We also evaluate the performance of OMG-LLaVA trained solely on the LLaVA-1.5 dataset on image-level benchmarks. When using the same LLM, OMG-LLaVA outperformed LLaVA 1.5 on MME, SEED-Bench, POPE, and AI2D by 41, 3.0, 3.0, and 5.1 points, respectively.

---

## 2. Performance of OMG-LLaVA (R#czBJ and R#Timw)

We have fixed some bugs and updated the performance of OMG-LLaVA, as shown in Table R1 (please refer to the submitted pdf). OMG-LLaVA achieves 78.0, 69.1 and 72.9 mIoU on refCOCO, refCOCO+ and refCOCOg benchmarks, outperforming LISA, PixelLM, and GSVA on the RES benchmarks. GLaMM uses the GranD pre-train dataset for pretraining. The GranD pretrain dataset contains over 11M images, **far exceeding** the GranD-f dataset used by OMG-LLaVA, which **only** includes 210K (0.21M) images. Nevertheless, OMG-LLaVA still surpasses GLaMM on the GCG benchmark by 0.6 CIDEr, 1.4 AP50, and 0.1 mIoU.

We have conducted experiments with OMG-LLaVA using stronger LLMs, such as Phi3 and Qwen2, and the results are presented in Table R3. With the stronger LLM Qwen2-7B, OMG-LLaVA performs better on pixel- and image-level benchmarks.

In addition, we will consider adding more segmentation and caption datasets for co-training, which will serve as our future work.

---

## 3. The contributions of OMG-LLaVA (R#czBJ and R#ck7P)

Our most significant contribution is introducing an elegant MLLM architecture that comprises only a perception model and an LLM, capable of understanding and reasoning at the image, object, and pixel level. This is a systematic contribution. To the best of our knowledge, we are the first open-source system to fulfill that goal in the MLLM domain using one image encoder, one image decoder, and one LLM. The frozen perception decoder can simultaneously encode visual prompts and decode [SEG] tokens. However, our experiments and those in LISA have shown that the frozen perception decoder cannot effectively decode [SEG] tokens. To better adapt the segmentation model to LLM, we propose the perception prior embedding module, which is the key component enabling our meta-architecture to work well.

In addition, compared with existing methods, OMG-LLaVA boasts an elegant model architecture and more streamlined workflows. For instance, in GLaMM, visual prompt encoding and segmentation mask decoding employ entirely different workflows and models. In OMG LLaVA, visual prompt encoding and segmentation mask decoding are unified into a single OMG Decoder to generate object-centric queries. Maintaining a consistent workflow while unifying all tasks is extremely challenging and currently difficult. For example, mask decoding and visual prompt encoding are only possible with LLMs that rely on other modules.  OMG-LLaVA significantly outperforms versatile MLLMs like LISA, PixelLM, and GLaMM in image-level tasks.

---

## 4. More ablation studies (R#Timw and R#UauW)

Following the reviewer's suggestion, we add additional ablation studies for better clarity. We have included ablation studies on the projector and perception prior embedding. The results are shown in Tables R4 and R5.

Table R4 presents the results of ablation studies on the projector. We observed a substantial performance degradation when pixel-centric and object-centric tokens shared a projector. This phenomenon can be attributed to the distinct semantic and perception information encoded in these two types of tokens. Sharing a projector results in detrimental interference between them. Furthermore, our experiments with additional cross-attention layers in the projector did not yield any performance benefits. For the object-centric tokens, specifically visual prompt tokens and object queries, sharing a projector resulted in marginal performance gains for both segmentation and visual prompt understanding tasks.

We have conducted ablation studies on different strategies for assigning perception embeddings to each pixel token, and the results are shown in Table R5. The best results are achieved by assigning the corresponding object query to each pixel using the SoftMax operator. The strategy of generating perception embeddings through L1 normalization causes performance degradation, but it still brought a significant performance improvement compared to not using the perception prior embedding strategy.

---

We will fix all the possible issues and improve the manuscript. To address all your concerns and questions, we prepared a comprehensive response, including additional experiments where necessary. If you have any further questions, please don't hesitate to interact with us.

Best regards

---

### Decision · Program_Chairs · 2024-09-25

**Decision:**

Accept (poster)

**Comment:**

This paper proposes OMG-LLaVA a framework for combining pixel-level vision understanding with reasoning abilities.  The paper received 1 strong accept, 1 accept, 1 borderline reject, and 1 borderline accept recommendations from reviewers. Positive points included the good motivation, clean and effective approach, generally clear writing, and good results. Negative points included lack of details for some approach components, limited technical novelty (especially by one reviewer), and missing ablation studies. Many of these concerns were adequately addressed by the rebuttal. There was an extensive post rebuttal discussion between the reviewers about the pros and cons of the paper. Overall, after carefully considering the paper, rebuttal, and discussions, the ACs feel that the paper makes a solid contribution, in which the positives outweigh the negatives, and recommend accept. It is recommended that the authors incorporate the rebuttal points into the final version.